# Hadaean to Palaeoarchaean stagnant-lid tectonics revealed by zircon magnetism

John A. Tarduno[1,2,3,4 ✉], Rory D. Cottrell[1], Richard K. Bono[5,6], Nicole Rayner[7], William J. Davis[7], Tinghong Zhou[1], Francis Nimmo[8], Axel Hofmann[9], Jaganmoy Jodder[9,10], Mauricio Ibañez-Mejia[11], Michael K. Watkeys[4], Hirokuni Oda[12] & Gautam Mitra[1]

Plate tectonics is a fundamental factor in the sustained habitability of Earth, but its time of onset is unknown, with ages ranging from the Hadaean to Proterozoic eons[1–3]. Plate motion is a key diagnostic to distinguish between plate and stagnant-lid tectonics, but palaeomagnetic tests have been thwarted because the planet's oldest extant rocks have been metamorphosed and/or deformed[4]. Herein, we report palaeointensity data from Hadaean-age to Mesoarchaean-age single detrital zircons bearing primary magnetite inclusions from the Barberton Greenstone Belt of South Africa[5]. These reveal a pattern of palaeointensities from the Eoarchaean (about 3.9 billion years ago (Ga)) to Mesoarchaean (about 3.3 Ga) eras that is nearly identical to that defined by primary magnetizations from the Jack Hills (JH; Western Australia)[6,7], further demonstrating the recording fidelity of select detrital zircons. Moreover, palaeofield values are nearly constant between about 3.9 Ga and about 3.4 Ga. This indicates unvarying latitudes, an observation distinct from plate tectonics of the past 600 million years (Myr) but predicted by stagnant-lid convection. If life originated by the Eoarchaean[8], and persisted to the occurrence of stromatolites half a billion years later[9], it did so when Earth was in a stagnant-lid regime, without plate-tectonics-driven geochemical cycling.

Ideally, the presence or absence of a mobile lithosphere can be tested using palaeomagnetism, but even the best preserved oldest rocks on Earth have experienced metamorphism, and this places severe restrictions on the type of magnetic carriers that might retain primary signals. Single-crystal palaeointensity (SCP), whereby single mineral crystals that contain magnetic inclusions capable of recording the ancient field are studied, provides an approach to see through this metamorphism[10]. This method has been applied to document the past strength of the geomagnetic field using progressively older Archaean rocks[11–13]. The results are consistent with the few available palaeointensity studies of whole rocks of low metamorphic grade[14,15] and/or those having signals dominated by silicate-hosted magnetizations[16,17]. Extant rocks older than approximately 3.45 billion years (Gyr) have generally been metamorphosed to amphibolite or higher grades, compromising any primary magnetic signals, but detrital single crystals found in younger sedimentary units have the potential to record even older geomagnetic fields[4]. The orientation of these detrital crystals at the time of their magnetization is unknown and, therefore, their magnetization direction does not constrain site latitude. The acquisition of palaeointensity data does not require knowledge of this orientation. SCP data can place bounds on motion because of the relationship between field strength and latitude, which is—in turn—a function of field morphology (Methods). Field-geometry constraints are unavailable before the

Neoarchaean[18], but recent studies indicate the lack of an inner core for the timespan considered here[19,20]. Modelling of the geodynamo without an inner core suggests a dipole-dominated field[20].

The only known detrital crystals that can be accurately dated, and that are able to provide constraints on lithospheric mobility spanning the multi-hundred-million-year timescales that typify plate-tectonic cycles, are detrital zircons bearing primary magnetic inclusions[4]. However, the magnetic measurement of zircons and interrogations of their magnetizations to determine whether they preserve primary signals are formidable technical challenges[4,6,7]. The first SCP measurements of zircons were from the JH Discovery site of Western Australia[6]. Exhaustive tests led to selection of only approximately 2% of the zircons separated from their host rock, and these yielded palaeointensity data suggesting the presence of a geodynamo in the Hadaean eon, approximately 4.2 Ga (ref. 6). This report was followed by a study showing how relatively young zircons successfully recorded the field[21] but also attempts to disprove the original findings[22,23]. Critiques have been systematically addressed[7,24–28] and, in so doing, further evidence on the preservation of primary magnetizations in zircons has been discovered. In particular, the presence of primary magnetite inclusions in select JH Discovery site zircons has been documented[7], consistent with magnetic unblocking temperature data presented in the first report on zircon magnetizations[6].

[1]Department of Earth and Environmental Sciences, University of Rochester, Rochester, NY, USA. [2]Department of Physics and Astronomy, University of Rochester, Rochester, NY, USA. [3]Laboratory for Laser Energetics, University of Rochester, Rochester, NY, USA. [4]Geological Sciences, University of KwaZulu-Natal, Durban, South Africa. [5]Geomagnetism Laboratory, University of Liverpool, Liverpool, UK. [6]Department of Earth, Ocean and Atmospheric Science, Florida State University, Tallahassee, FL, USA. [7]Natural Resources Canada, Geological Survey of Canada, Ottawa, Ontario, Canada. [8]Department of Earth and Planetary Sciences, University of California, Santa Cruz, Santa Cruz, CA, USA. [9]Department of Geology, University of Johannesburg, Auckland Park, South Africa. [10]Evolutionary Studies Institute, University of the Witwatersrand, Wits, South Africa. [11]Department of Geosciences, University of Arizona, Tucson, AZ, USA. [12]Research Institute of Geology and Geoinformation, Geological Survey of Japan, National Institute of Advanced Industrial Science and Technology (AIST), Tsukuba, Japan. ✉e-mail: john.tarduno@rochester.edu

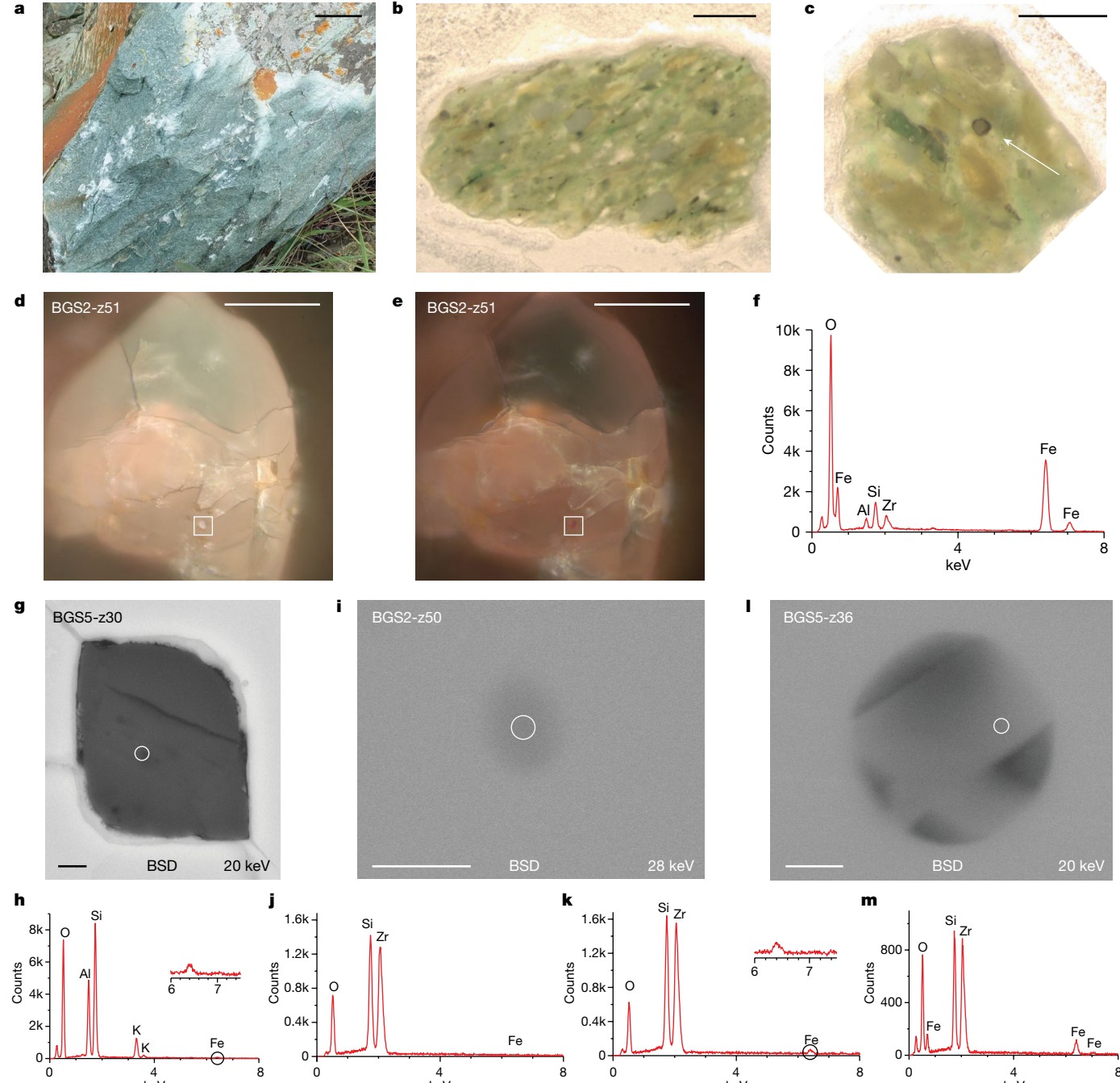

**Fig. 1 | BGS, detrital zircons and Fe-bearing inclusions. a**, Field photo of BGS. Scale bar, 5 cm. **b**, Example section of BGS approximately 300 µm thick. Scale bar, 1 mm. **c**, Example of detrital zircon (arrow) in BGS matrix. Scale bar, 1 mm. **d**,**e**, Reflected-light images (100×) of zircon BGS2-z51, with the boxes highlighting Fe-oxide inclusion (magnetite) at 0° (**d**) and 90° (**e**) polarization. Scale bars, 50 µm. **f**, SEM EDS spectra of Fe inclusion highlighted in **d** and **e** (see Extended Data Fig. 2 for further SEM images and EDS analyses). **g**, SEM BSD image of silicate inclusion in zircon BGS5-z30 with EDS analysis location identified by the circle (see Extended Data Fig. 1 for further SEM images and EDS spectra). Scale bar, 1 µm. **h**, EDS spectra of **g** with Fe signal highlighted. **i**, SEM BSD image of Fe-oxide inclusion (magnetite) in zircon BGS2-z50 with EDS spectra analysis location highlighted by the circle. Scale bar, 200 nm. **j**,**k**, EDS analyses of **i**. An Fe signal was not observed using a 20-keV beam (**j**) but was detected using a 28-keV beam (**k**), emphasizing that the inclusion is at depth (>4 µm). (Methods. Corresponding reflected-light images showing inclusion extinction are in Extended Data Fig. 2.) **l**, SEM BSD image of Fe-oxide particles in zircon BGS5-z36 with EDS location highlighted by the circle (further reflected-light and SEM images are shown in Extended Data Fig. 2). Scale bar, 200 nm. **m**, EDS spectra of **l**.

The distinctive time sequence defined by the JH palaeointensity values[6,7] is one of several lines of evidence indicating primary magnetizations rather than magnetic resetting ('Evidence for primary magnetite inclusions and magnetizations in JH zircons' section in Methods). However, before using the JH palaeointensity history to assess plate mobility, we have sought another record to further test its fidelity. Towards that goal, we sampled the Barberton Green Sandstone (BGS) of the Barberton Greenstone Belt, Kaapvaal Craton, South Africa (Fig. 1a–c), which has yielded detrital zircons ranging in age from 3.3 to 4.15 Gyr (refs. 5,29). The BGS is less deformed and of lower metamorphic grade than the JH metaconglomerate zircon host rocks, with peak temperatures of ≤350 °C (ref. 30). Palaeomagnetic conglomerate tests from the Barberton Greenstone Belt indicate the preservation of a high unblocking temperature primary magnetization[14,15].

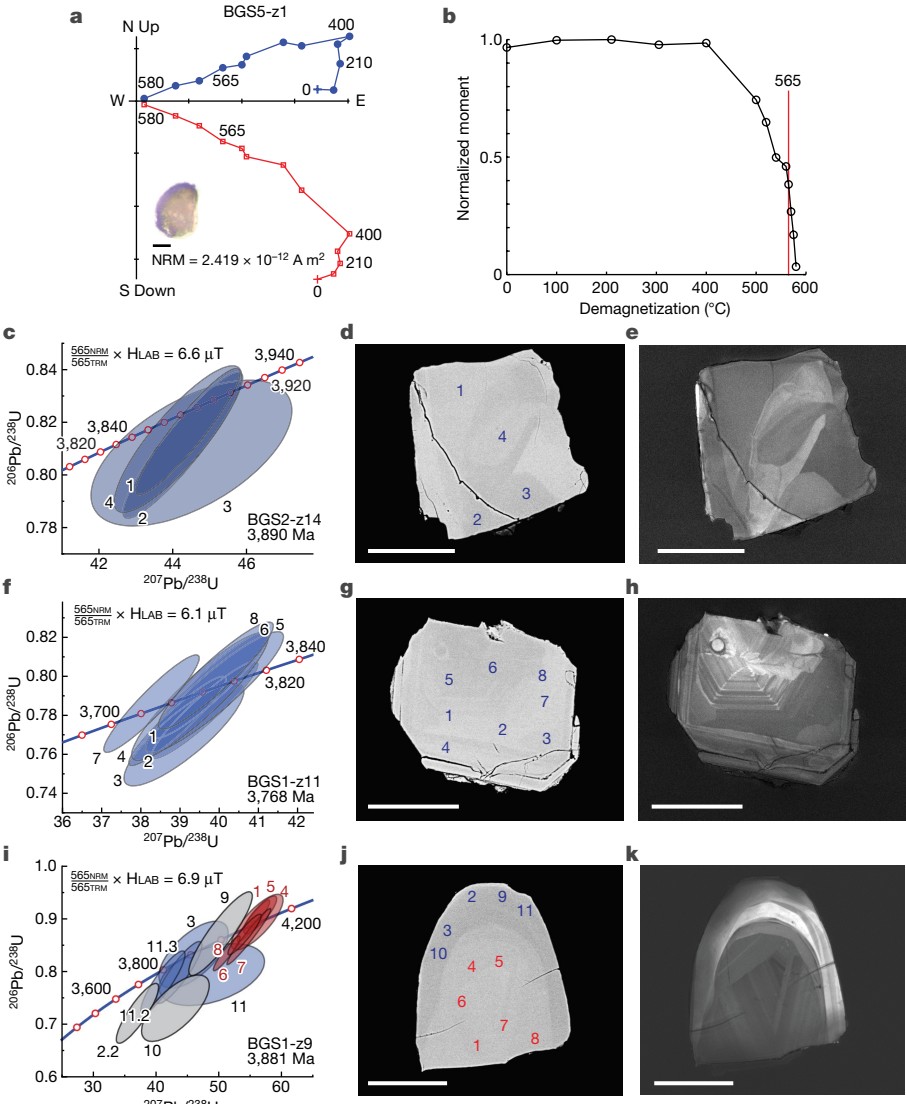

**Fig. 2 | Palaeomagnetic thermal demagnetization, palaeointensity determinations and SHRIMP age data from individual BGS zircons.**
**a**, Orthogonal vector plot of thermal demagnetization of unoriented zircon BGZ5-z1 (shown as inset). Temperatures shown are in °C. Red, vertical projection of the magnetization; blue, horizontal projection. Scale bar, 50 μm. **b**, NRM moment after thermal demagnetization (**a**) normalized to its undemagnetized value plotted versus demagnetization temperature. Value after heating at 565 °C highlighted (vertical red line). **c**, Concordia diagram showing SHRIMP geochronological analyses (uncertainty ellipses are 2σ) and palaeointensity value for zircon BGS2-z14. **d**, Corresponding backscattered scanning electron microscope image with analysed spots labelled. **e**, Corresponding cathodoluminescence image. **f**–**h**, Analyses as shown in **c**–**e** for zircon BGS1-z11. **i**–**k**, Analyses as shown in **c**–**e** for zircon BGS1-z9. Red, analyses from core; blue, analyses from rim; grey, excluded from calculation of mean age. Scale bars, 50 μm.

We separated zircons using a non-magnetic heavy-mineral technique (Methods). Reflected-light, scanning electron microscopy (SEM) and energy-dispersive X-ray spectroscopy (EDS; Methods) images show a diverse population of single-mineral inclusions and multiphase melt inclusions in the zircons, including apatite (Extended Data Fig. 1a–c), quartz (Extended Data Fig. 1d–g), feldspar (Fig. 1g,h and Extended Data Figs. 1h–n and 2a–f,h) and iron oxide (Fig. 1d–m and Extended Data Fig. 2f–j). Fractures record stress concentration around inclusions, providing further evidence that the inclusions are primary ('Microtectonic analyses of zircons' section in Methods and Extended Data Figs. 1 and 2). The iron-oxide inclusions show a variety of occurrences similar to those previously reported for JH zircons[6,7]. These include: (1) relatively large (>1 μm) particles (Fig. 1d,e and Extended Data Fig. 2f), (2) isolated iron-oxide particles at depth (Fig. 1i–k) and (3) Fe-oxide particles that crystallized in small (<1 μm) pockets (Fig. 1l,m). We also detected Fe that could be in the form of inclusions within larger silicates (Fig. 1g,h,

Extended Data Fig. 1h–n and 'Reflected-light, SEM, palaeomagnetic and U-Pb analyses' section in Methods). For potential inclusions inside melt inclusions and/or silicates, we cannot assign an Fe-oxide type, but for others (that is, Fig. 1d–f,i–k and Extended Data Fig. 2g,i,j), reflected-light microscopy and EDS observations together indicate a magnetite composition ('Reflected-light, SEM, palaeomagnetic and U-Pb analyses' section in Methods). The Fe-oxide particles detected (for example, Fig. 1i,l) include those in the size/shape range to have single-domain or stable single-vortex characteristics, with relaxation times of billions of years required to preserve primary signals.

For palaeointensity analysis, we adopt zircon selection criteria similar to those in our studies of JH zircons (Methods). Salient aspects of the analysis include the use of a $CO_2$ laser for demagnetization, allowing heating durations at least an order of magnitude less than those typical of conventional palaeomagnetic ovens, limiting laboratory alteration, and measurement with a WSGI ultrasensitive three-component

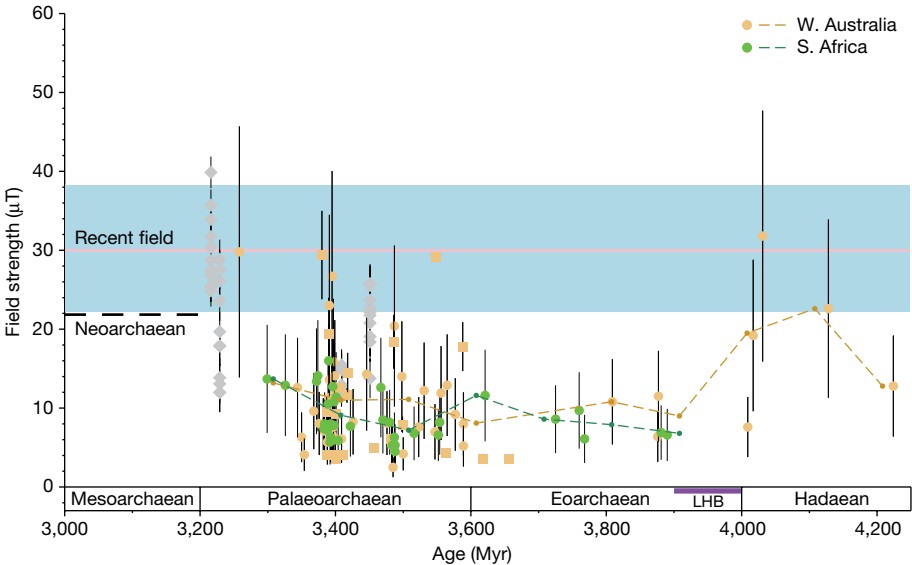

**Fig. 3 | Palaeointensity history from BGS (South Africa) and JH (Western Australia) zircons.** Zircon palaeointensity results: green circles, 565 °C palaeointensity determinations (this study); yellow from the JH (boxes, Thellier–Coe palaeointensity results; circles, 565 °C palaeointensity determinations)[6,7]. Green and yellow dashed lines: 100-Myr running average of zircon palaeointensity results from the BGS and JH, respectively. Other single-silicate palaeointensity results from extant igneous rocks shown as grey diamonds[12,13]. Recent field: pink solid line is mean and standard deviation (blue shaded region) from a bootstrap resampling of data[6] from the past 800 thousand years, set to the palaeolatitude of the Mesoarchaean data[12]. Neoarchaean field strength (dashed black line) based on mean of select time-averaged palaeointensity results[16,17]. LHB, Late Heavy Bombardment. All data are above threshold for geomagnetic field presence based on external field imparted by the solar wind[4]. Near-constant palaeointensity values between approximately 3.9 Ga and approximately 3.4 Ga indicate palaeolatitude stasis of the recording sites (see text).

direct-current superconducting quantum interference device (SQUID) magnetometer at the University of Rochester (Methods). The natural remanent magnetization (NRM) intensity of samples selected for analysis range from $9.4 \times 10^{-13}$ A m² to $2.5 \times 10^{-12}$ A m², representing approximately 13% of the more than 1,000 zircons separated ('Reflected-light, SEM, palaeomagnetic and U-Pb analyses' section in Methods). Demagnetization experiments yield results typical of magnetite, with complete unblocking at about 580 °C (Fig. 2a,b). On the basis of these results, we use the 565 °C palaeointensity method[6,7], developed to further limit laboratory alteration and to retrieve field-strength estimates to compare with the JH record (Methods).

After magnetic analysis and the application of palaeointensity selection criteria (Methods), we analysed zircons using the Geological Survey of Canada SHRIMP II. Thirty-five zircons passed the selection criteria, representing approximately 3.5% of those separated (Methods), and their $^{207}Pb/^{206}Pb$ ages are considered further (Supplementary Table 1 and Supplementary Information). These yield Eoarchaean (Fig. 2) to Palaeoarchaean (Extended Data Fig. 3) concordant ages. Several zircons have older Hadaean-age cores (for example, Fig. 2i–k), extending back in time to 4.2 Ga (or older) (Extended Data Fig. 3j–l). However, these Hadaean zircons also have late Hadaean to early Eoarchaean, and sometimes younger, zircon rims (for example, Extended Data Fig. 3j–l). The late Hadaean to early Eoarchaean ages are intriguing because of their coincidence in time with the Late Heavy Bombardment[31]. In cases of evidence for growth of zircon rims, we assign the rim age with the time of magnetization.

Using this approach, we obtain a time sequence of palaeointensity values ranging from approximately 3.9 Ga to approximately 3.3 Ga (Fig. 3 and Extended Data Table 1). We find that we cannot reject the null hypothesis that these BGS data and those from the JH having ages between 3.9 and 3.4 Gyr sample the same underlying field distribution at the 95% confidence level using Welch's *t*, Kolmogorov–Smirnov (KS) and Mann–Whitney *U* tests ('Statistical analysis of BGS and JH zircon palaeointensity data' section in Methods and Extended Data Table 2). For times younger than 3.4 Ga, however, Welch's *t*-test suggests that the

BGS and JH palaeointensities begin to sample different distributions ('Statistical analysis of BGS and JH zircon palaeointensity data' section in Methods and Extended Data Fig. 4a). Although the data between 3.9 and 3.4 Ga are indistinguishable, the variance of the JH data is higher than that in the BGS data, which could reflect either a difference in recording fidelity and/or the time over which the field is recorded ('Statistical analysis of BGS and JH zircon palaeointensity data' section in Methods). Furthermore, the data are not evenly distributed in time, suggesting that only the broad character of the field over long timescales can be retrieved. Accordingly, we calculate 100-Myr time averages (Extended Data Table 3) to ensure that the time-averaged field needed to recover a dipole dynamo signal (palaeomagnetic dipole moments (PDMs), equations (1) and (2) in Methods) are represented. We find that the JH palaeointensity history based on the 100-Myr time window averaging (Extended Data Fig. 4b) predicts the BGS data (Extended Data Fig. 4c) over this same time range (KS test *P* = 0.32, above a significance level of 0.05; 'Statistical analysis of BGS and JH zircon palaeointensity data' section in Methods and Extended Data Fig. 4d–f). This similarity from widely separated sites with different Palaeoarchaean to recent geologic histories strongly support the interpretation that select JH and BGS zircons preserve a primary global signal of the geodynamo (also see the 'Reflected-light, SEM, palaeomagnetic and U-Pb analyses' section in Methods). But the unchanging 3.9–3.4-Ga palaeointensity values also indicate constant site latitudes.

We next test whether constant latitudes could arise from a modern style of plate tectonics by synthetically sampling continental areas and tracing their motion over the past 600 Myr using plate reconstructions[32] ('Plate-motion analysis' section in Methods and Extended Data Figs. 5 and 6). Geochemical data from JH zircons support an andesitic source, interpreted to record a modern arc-type setting[33], whereas those from the BGS support more heterogeneous crustal sources, including those distinct from that of the JH[29]. On this basis, we consider the case that the BGS and JH zircons were formed on at least two notional plates. However, no examples of two plates at constant latitudes could be found in the past 600 Myr of plate tectonic history.

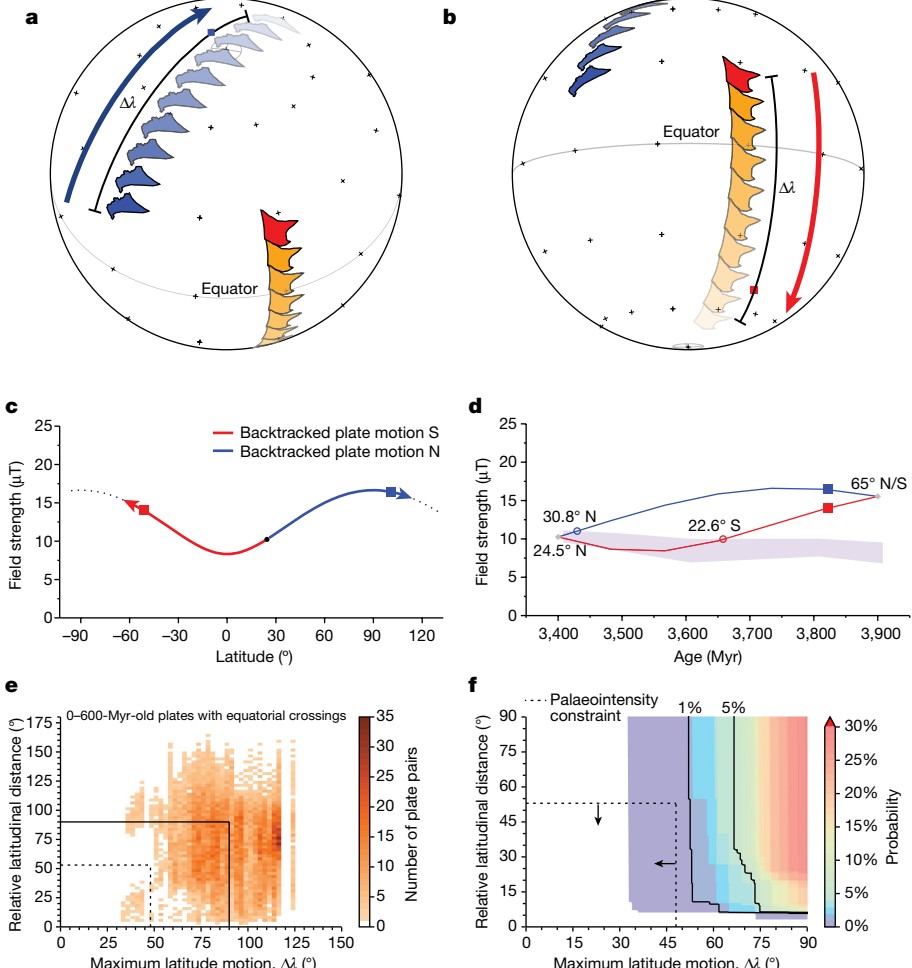

**Fig. 4 | Hypothetical latitudinal motions typical of 0–600 Myr plate tectonics. a**, Two hypothetical continents, one backtracked north (blue) and the other south (red, orange), from a starting latitude of 24.5° corresponding to the 3.4–3.45-Ga palaeolatitude of the BGS[13] (hemisphere is unknown and arbitrarily set as N). Northward motion (blue arrow) and maximum latitudinal motion (Δλ) highlighted. Blue square represents the median Δλ value observed for continental plates of the past 600 Myr. **b**, Same as **a** but the continent backtracked south is highlighted (in red and red arrow). Red square represents the median maximum latitudinal value observed for continental plates of the past 600 Myr. **c**, Dipole relationship between field intensity and latitude, set to the palaeolatitude of the BGS at 3.4–3.45 Ga (hemisphere is unknown and arbitrarily set as N), and the BGS/JH field strength value. Squares as in **a** and **b**. **d**, BGS and JH palaeointensity

data (W. Australia + S. Africa) shown as the standard error of the 100-Myr bin mean (violet band) with predicted latitude history for a site moving northward (**a**) or southward (**b**). Squares as in **a** and **b**. Open circles are the values for which the back-tracked palaeolatitude began to differ from the observations. **e**, 2D histogram of the relative latitudinal distance and maximum latitude motion (Δλ) characteristics of equatorial crossing plates of the past 600 Myr. Colour scale shows the number of unique plate pairs that exhibit motion as in **b**. Data are grouped into 2.5° bins. Dashed lines are the BGS/JH palaeointensity constraints (**d**; see 'Statistical analysis of BGS and JH zircon palaeointensity data' section in Methods). **f**, Expanded view of **e**, shown as the probability of sampling a pair of equatorial crossing plates with these relative latitudinal distance and Δλ characteristics. Dashed lines and arrows are the BGS/JH palaeointensity constraints.

We next investigate the resolution of the BGS/JH palaeointensity data in constraining latitudinal motion by considering hypothetical histories of plates moving back in time from 3.4 to 3.9 Ga ('Plate-motion analysis' section in Methods). We start at a palaeolatitude of approximately 24.5° constrained by the palaeomagnetism of extant Palaeoarchaean BGS rocks[13] (but arbitrarily set to the Northern Hemisphere) and allow the plates to traverse 90° of latitude, one backtracking south, the other north (Fig. 4a–d). The median maximum latitudinal travel for plates of the past 600 Myr is approximately 76°, different from the history inferred from the BGS/JH data (Fig. 4d).

The dipole relationship of field with latitude for a site originating at 24.5° creates an asymmetry in the resolving power of intensity data for a site backtracked to the north versus one backtracked to the south (Fig. 4c,d). Specifically, if we consider a site at 24.5° N, the BGS/JH intensity data are inconsistent with more than about 5° of latitudinal motion backtracked to the north (Fig. 4d). However, the same site back-tracked to the south could have crossed the equator and have motion

compatible with the data until it passed a latitude of 22.6° S, for a total latitudinal travel of approximately 47° (Fig. 4d). There is no example of a plate pair in the past 600 Myr that meets the requirements of the northward backtracked case. The probability of a plate pair meeting requirements of the southward backtracked possibility is <1% (Fig. 4e,f and 'Plate-motion analysis' section in Methods). Palaeolongitude is not directly constrained by our data, but our analyses show that plates with dominantly longitudinal motion are rare over the past 600 Myr ('Plate-motion analysis' section in Methods). Moreover, because all plates show substantial latitudinal motion, our comparative tests inherently include these cases.

## Discussion

The stark differences between the palaeolatitude history inferred for the BGS and JH sites between 3.9 and 3.4 Ga and Phanerozoic plate tectonics, highlighted by the comparisons above, provide evidence for a

non-mobile lithosphere. Together with indications of the growth of continental crust from zircons[33,34], this indicates a regime of stagnant-lid tectonics[35] for Earth in Palaeoarchaean to Hadaean times, with plate tectonics, as defined by large-scale horizontal motions, possibly commencing after 3.4 Ga ('Prior geologic models and attendant implications' section in Methods). Processes to cool the mantle during the stagnant-lid regime should have included plumes and/or heat pipes[36]. We note that there is geological evidence in the Eoarchaean Itsaq Gneiss Complex of southwestern Greenland for crustal shortening[37]. However, our data suggest that this shortening and, more generally, the crustal recycling needed to generate continental crust[33,34] must have been spatially localized without extensive deep subduction creating highly mobile plates. Similarly, the lack of a mobile lithosphere indicates that potential subduction initiated by impacts[38] did not lead to a global system of plates and plate boundaries similar to that of today.

The Eoarchaean to Palaeoarchaean eras may be unique in Earth history because our data indicate that changes in geodynamo efficiency, which can otherwise dominate field strength, were minor ('Palaeointensity variations and dynamo efficiency' section in Methods). Thus, palaeointensity can provide first-order bounds on the locations of the first sites of crustal generation. In particular, we note that the palaeolatitude/latitude stasis, together with the latitudinal constraints at 3.4–3.45 Ga from extant rocks[13], indicate that the sites of crustal generation producing the BGS and JH zircons were at low latitudes from the Eoarchaean to Palaeoarchaean eras, and probably since the Hadaean eon.

The geochemical cycling provided by plate tectonics is recognized as being a key factor in sustaining habitability of Earth. However, it has been less clear whether plate tectonics was required for the origin and early viability of life. Our new data suggest that, if life persisted throughout the Eoarchaean to Palaeoarchaean, it did so while Earth was in the stagnant-lid convective regime, demonstrating more broadly that a modern style of global plate tectonics is not a requirement for life on terrestrial planets during their first billion years. Moreover, the latitudinal history of the BGS and JH zircons indicates stability of the Earth relative to the spin axis, which is more likely in a planet without deep (>660 km) subduction[39]. The lack of large, rapid changes in environmental conditions induced by true polar wander likely fostered survival of nascent life on our planet.

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

## Methods

### Sample preparation and selection

Zircons are more dispersed in the BGS relative to zircons in the JH Discovery site samples studied by Tarduno et al.[6,7] and, therefore, separations were performed at the University of Rochester using non-magnetic gravimetric methods. Methylene iodide was the heavy liquid used and zircons were handpicked from the separate. Zircon selection follows the methods of Tarduno et al.[6,7], with some modifications. The BGS zircons isolated are generally slightly smaller than those studied from the JH Discovery site and, therefore, we focus on zircons larger than approximately 70 μm (versus 150 μm). The NRM intensity selection cutoff for analysis (applied to ensure that demagnetized values can be accurately measured) was slightly lower for the BGS zircons (approximately $9 \times 10^{-13}$ A m$^2$ versus approximately $1 \times 10^{-12}$ A m$^2$).

Viable zircons were collected at three sites (BGS1, BGS2 and BGS5). The separations yielded >1,000 zircons, but only about 500 met the size requirements. Of these, approximately 300 crystals could be isolated that were single zircons without attached non-zircon grains and that were clear of obvious surface alteration. One hundred and thirty zircons had NRM values $\geq 9 \times 10^{-13}$ A m$^2$. Sixty-five of these crystals without visible fractures or large inclusions (that might be multidomain oxides) were used for the 565 °C palaeointensity experiments. Seven zircons were rejected before experiments because of non-reproducibility of initial NRM intensity measurements; the initial value was likely contaminated by a viscous remanent magnetization. A further five zircons failed because of a large intensity drop after the first thermal treatment, again suggesting a viscous remanent magnetization. Six zircons failed palaeointensity checks (they did not lose NRM on heating or did not gain thermoremanent magnetization (TRM) or failed the multidomain-tail check). Forty-seven zircons meeting palaeointensity criteria as established by Tarduno et al.[6] were subject to SEM and cathodoluminescence analyses and 12 were rejected because of metamictization or other evidence of internal grain disturbance. The 35 selected zircons represent a success rate of at least approximately 3.5% from those isolated from the methylene iodide heavy liquid separate. Although techniques differ, this success rate is higher than that of the study of JH zircons (<2%; Tarduno et al.[6]), consistent with a better preservation of the BGS zircons.

### Optical and scanning electron microscopy

Optical microscopy studies used a Nikon Eclipse LV100POL microscope with both transmitted and reflected-light capabilities, a maximum of 1,000× magnification and a SPOT Insight 4-MP CCD colour digital camera assembly. SEM analyses were also conducted at the University of Rochester using a Zeiss Auriga SEM. For subsurface inclusions, we estimated depth using Electron Flight Simulator Version 3.1E software.

### Palaeointensity experiments

Procedures follow those of Tarduno et al.[6,7] and are outlined here. All experiments were conducted in the shielded room (ambient field < 200 nT) of the Paleomagnetism Laboratory at the University of Rochester. The 565 °C palaeointensity method[6] was used to derive palaeofield strength estimates while limiting the effects of laboratory-induced alteration by reducing the number of heatings. A temperature of 565 °C is chosen for comparison with the JH palaeointensity dataset, which also used this temperature, and to represent blocking temperatures best reflecting single-domain-like magnetite inclusion carriers, while also retaining a measurable demagnetized value. The demonstration that 565 °C falls within the interval of sharp magnetic unblocking (Fig. 2b) supports this choice. In the 565 °C palaeointensity approach, the NRM is first measured, and then the zircon was gradually heated in field-free space with a Synrad v20 $CO_2$ laser[12]. Specifically, the temperature was held constant for 1 min at 100-°C temperature steps to 500 °C. Next, the temperature was increased to 565 °C for an extra 1 min and then

allowed to cool for 3–5 min. Thereafter, a second heating to 565 °C was conducted in the presence of a field. An applied field of 15 μT was used in all field-on treatments. A multidomain check was performed last by reheating the zircon sample to 565 °C in the absence of a field. We tabulate the calculated angle between the TRM vector ($565_{on}$-$565_{off}$; Extended Data Table 1) and the applied field. Differences from zero may record small error contributions related to the alignment of the laser beam and a zircon in these challenging experiments and/or an anisotropy of the collection of magnetic particles in a given zircon.

### Relating palaeointensity to palaeolatitude

The geomagnetic field can be as described by the scalar potential $\Psi_m(r, \theta, \phi, t)$:

$$
\begin{aligned}
\Psi_m(r, \theta, \phi, t) \\
= \frac{r_e}{\mu_0} \sum_{l=1}^{\infty} \sum_{m=0}^{l} \left(\frac{r_e}{r}\right)^{l+1} P_l^m \cos\theta \left[g_l^m(t)\cos m\phi + h_l^m(t)\sin m\phi\right]
\end{aligned}
\tag{1}
$$

in which $P_l^m$ are partially normalized Schmidt functions, $l$ and $m$ are spherical harmonic degree and order, respectively, $r_e$ is the radius of Earth and the Gauss coefficients $g_l^m(t)$ and $h_l^m(t)$ describe the spatially and time-varying fields. In the case of a time-averaged field needed to make conclusions on the geodynamo[10], the geocentric axial dipole ($g_1^0$) is represented by a PDM and field strength $B$ is related to palaeolatitude ($\lambda$) by:

$$
B = \frac{\mu_0 \text{PDM}}{4\pi r^3} \sqrt{1 + 3\sin^2\lambda}
\tag{2}
$$

### Geochronology

Standard SHRIMP U-Pb instrumental setup[40] and procedure for U-Pb calibration by reference materials[41] were followed. Two 1-inch epoxy mounts (IP985 and IP987) were built containing the JH and BGS zircons as well as zircon reference materials z6266 with an accepted $^{206}$Pb/$^{238}$U age of 559 Myr[41] and z1242 with an accepted $^{207}$Pb/$^{206}$Pb age of 2,679.6 ± 0.2 Myr[42]. Polishing with diamond suspension exposed the internal structure of the zircon grains. After coating with 10 nA of gold, the grains were imaged in cathodoluminescence (CL) and backscattered electron detector (BSD) mode using a MIRA3 TESCAN field emission scanning electron microscope. In order to target pristine areas of zircon grains and avoid cracks and alteration, a $^{16}$O- ion beam 13 μm in diameter and with an average beam current of 1 nA was used. The analytical runtable consisted of 11 masses including background with species of Hf, Yb, Zr, Pb, Th, U analysed over six scans. Data reduction used SQUID3 (ref. 43) (note that the name of this software has no relation with SQUID magnetometers; SQUID3 refers to the name of the software and it is not an acronym). Steiger and Jäger[44] decay constants were used. Two analytical sessions were carried out on each of the two epoxy mounts. The 1σ external error for the $^{206}$Pb/$^{238}$U calibration relative to reference material z6266 for sessions IP985_1 and IP985_2 was ±0.88% or ±1.01, respectively. The 1σ external error for the $^{206}$Pb/$^{238}$U calibration for sessions IP987_2 and IP987_3 was ±0.75% or 1.07%, respectively. These errors are also specified in the footnote to the data table (Supplementary Table 1). The requirement for a correction due to instrumental mass fractionation of the Pb isotopes was assessed through replicate analyses of reference material z1242. The measured weighted mean $^{207}$Pb/$^{206}$Pb age of those analyses are reported for each session in the footnotes of Supplementary Table 1. No fractionation correction was applied to the Pb- isotope data except for session IP987_2, in which a correction of −0.64% was applied. Concordia plots were generated, and weighted means calculated using Isoplot v. 4.15 (ref. 45). The uncertainties for the weighted mean ages reported in the text and Supplementary Materials are at the 95% confidence level.

**Evidence for primary magnetite inclusions and magnetizations in JH zircons.** Palaeomagnetic, reflected-light microscope, electron microscope, microtectonic, geochemical and palaeointensity data indicate the presence of primary magnetite inclusions in select JH zircons and that these have primary magnetic signals[6,7]. Specifically, zircon microconglomerate tests and the identification of distinct secondary components of magnetization separate from the primary characteristic magnetization exclude magnetic resetting after incorporation of the zircons into the JH host metasediment (figure 2 of ref. 7). [7]Li profiling data (for example, figure 6 of ref. 7) argues against thermal resetting of Hadaean and Eoarchaean data, whereas reflected-light microscopy, microtectonic analysis, SEM/EDS, focused ion beam and NanoMOKE investigations document the presence of primary magnetite inclusions (figures 3–5 of ref. 7). Pb-Pb screening of data, documented in refs. 6,7 (figure 7 of ref. 7) argues against a magnetic resetting age older than the age of incorporation into the host sediment in the selected zircons. This is further enforced by the distinct change in palaeointensity data, with values from late Hadaean zircons being higher than those of Palaeoarchaean to Eoarchaean age, inconsistent with Palaeoarchaean magnetic resetting. Reference 6 notes that iron oxyhydroxides can be found in JH zircons that are likely related to weathering, but these cannot be the carrier of the magnetic remanence used for palaeointensity determination. Furthermore, an assemblage of magnetite inclusions is needed to account for the magnetizations observed from the zircon, satisfying Maxwell–Boltzmann statistical limits on magnetic recording[7]. Reference 7 also describes why remagnetization scenarios calling on the neoformation of magnetic grains represent particles too small or too few in number to record stable magnetizations. Reference 7 also notes that the association of magnetic particles with dislocations does not mean that the particles are secondary[22]; instead, this is expected given the deformation history of the JH conglomerate.

Although the zircon studied would not pass our selection criteria, atom probe tomography[46] data on a JH zircon illustrate differences in the high unblocking temperature magnetic primary remanences isolated in refs. 6,7 and other potential Fe-bearing minerals. Reference 46 interprets quantum diamond microscope data from ref. 22 on a single zircon as indicating Fe-bearing zones and links those to the carriers of natural magnetic remanence. Atom probe tomography data from one zone within that zircon is further interpreted to contain Fe nanoclusters (which are far too small to carry remanent magnetizations) and a maximum age of approximately 1.4 Gyr is assigned; this is interpreted as a natural remanent magnetization age. A quantum diamond microscope does not have the sensitivity to measure the natural magnetic remanence of JH zircons. Reference 7 explains that the authors instead measured the magnetization of a laboratory-induced isothermal remanence magnetization (0.25 T, >4,000× present-day field; ref. 22); this strong field can enhance the magnetization of Fe oxides/oxyhydroxides that do not contribute to the primary remanence. The zircon microconglomerate tests[6,7] discussed above supersede older tests and document that the high unblocking temperature magnetization in JH zircons must be older than the approximately 3-Gyr depositional age of the JH conglomerate. Therefore, the hypothetical remanent magnetization inferred in ref. 46 with an assigned age of approximately 1.4 Gyr cannot be the key high unblocking temperature component of JH magnetization[6,7].

**Microtectonic analyses of zircons.** Samples for this study were collected from the lower approximately 1 m of the approximately 3–5-m-thick BGS. Our samples appear distinct from those of the lowest 20 cm of the BGS, which show extensive shear zones in thin section[47]. In thin section, our BGS samples used for zircon separation have a greenschist-grade anastomosing foliation in the generally fine-grained matrix. The foliation is cut by several conjugate through-going fractures. The competency difference between the competent zircon grains and the weaker surrounding matrix could result in stress buildups at the grain boundaries

that might generate cracks within the zircon grains. Below, reference is made to specific zircons studied by reflected-light and scanning electron microscopy.

*SEM sample BGS2-z51.* The fractures appear to be in three main systems (Extended Data Fig. 2a). The 'vertical' or N–S fractures are parallel to the *c* axis (crystallographic axis based on geometry). The 'east-dipping' and 'west-dipping' sets appear to be part of the same (112) crystallographic system as close-packed planes (for example, noted by Reddy et al.[48]). The melt inclusions (Extended Data Fig. 2b,f,d) are interesting because that in panel d does not have a fracture going through it, further indicating that the inclusions are not being formed by material diffusing in through the fractures. Instead, these patterns suggest that the inclusions are primary and many of the fractures are localized on the inclusions. The competency difference between the host crystal and the inclusions likely result in stress concentrations at the edges of the inclusions (depending on their composition). The fractures would tend to propagate along close-packed crystallographic planes (that is, the (112) planes).

*Reflected-light sample BGS5-z30*, Extended Data Fig. 1h,i. From the form of the crystal, the *c* axis is approximately vertical. There are two sets of conjugate fractures with the acute angle between them facing approximately E–W (they form the V pattern in the upper part of the grain), which would suggest an E–W compression direction when those fractures formed. The E–W fracture on either side of the 'football-shaped' inclusion probably formed during the same compression. The fractures coming off the tapered ends of the 'football-shaped' inclusion are likely guided by the inclusion itself. The tapered ends of an inclusion have the smallest radius of curvature and, therefore, generate the highest stress concentrations, causing the fracture to propagate out from the tapered tip. Those fractures do not appear to be guided by the crystal structure of the zircon—if the fractures had propagated farther, they might have reoriented to follow a crystallographic close-packed plane. There is a possible fracture within the inclusion (Extended Data Fig. 1j) that could reflect propagation from a stress concentration outside the inclusion.

*Reflected-light sample BGS2-z39*, Extended Data Fig. 1. Although the fractures look random at first glance (Extended Data Fig. 1d), a few fractures seem to be parallel to the *c* axis or 001 (along the length of the crystal), a set is perpendicular to the *c* axis (on the bottom of the crystal) and there is a pair of conjugate sets (perhaps the (112) set) indicating compression approximately in the E–W direction. The globular form of the quartz inclusion highlighted in Extended Data Fig. 1e,f suggests that it is a melt inclusion. The adjacent fracture does not pass through the inclusion. The fractures likely originate at the inclusion boundary as a result of stress concentration created by the inclusion. The fracture below the inclusion is clearly following a crystallographic plane parallel to the *c* axis.

*SEM sample BGS2-z36*, Fig. 1l and Extended Data Fig. 2k–l. The inclusion is not connected to fractures in the 2D view available. Lattice diffusion (Nabarro–Herring creep) requires temperatures greater than half the melting temperature. In the case of zircon, that would be approximately 900 °C, unless the crystal structure was sufficiently disturbed that the lattice was more open. There is no evidence of metamictization that might signal such an open structure. The distance from the edge of the crystal to the inclusion (in 2D) is approximately 2 μm and it is unlikely that the inclusion and interior Fe particles could form by diffusion through the crystal lattice. Instead, it is likely that this is a melt inclusion formed during the original formation of the crystal (at temperatures >1,000 °C).

**Reflected-light, SEM, palaeomagnetic and U-Pb analyses.** Magnetic inclusions on the tens to hundreds of nanometres scale are well established within silicates (for example, Tarduno et al.[10]); trace Fe signals could reflect such inclusions within the zircons, but further work is needed to distinguish these from Fe intrinsic to the crystals in the case

of feldspar. When viewed under reflected light, we note that isolated inclusions often show extinction after rotation from 0° to 90° polarization angles. For those shallow enough in the crystal to be sampled by SEM EDS analysis, the presence of Fe together with extinction provides evidence for magnetite[7].

We emphasize that the zircons in our study were selected using robust protocols for the isolation of primary magnetic recorders, established in the first palaeomagnetic study of zircons[6]. Our NRM and demagnetization data, together with our transmitted/reflected-light and SEM observations, show that a claim that BGS zircons "contain virtually no ferromagnetic minerals"[49] is incorrect.

To demonstrate the robustness of the isotopic system of individual zircon grains, replicate analyses were carried out wherever possible. Zircon grains with reproducible Pb-Pb ages are inferred to be closed systems that have not lost Pb due to diffusion, recrystallization or other thermal processes. When the probability of fit of the weighted mean age is greater than 0.05, this age is considered to be the magnetization age. In cases in which zircon overgrowths are present, we assign the youngest robust age with the time of magnetization. A description of the results of each individual grain is available in the Supplementary Information. Overall, we note that our BGS zircons lack 3.4 rims. This is consistent with our interpretation that these zircons have not experienced a 3.4-Ga high-temperature event that might otherwise have affected the magnetic history they record.

**Statistical analysis of BGS and JH zircon palaeointensity data.** To compare the JH and BGS zircon palaeointensity data, we first examine the following hypotheses: $H_0$, the JH and BGS data sample the same underlying field distribution, designated as the null hypothesis; and $H_1$, the JH and BGS data sample different underlying field distributions, designated as the alternative hypothesis. We focus on three statistical tests: the Welch version of the Student $t$-test (Welch[50]), the two-sample KS test[51] and the Mann–Whitney $U$ test[52]. The Student $t$-test differs from the KS and Mann–Whitney $U$ tests in the assumption that each sample is normally distributed with equal variance. Given the small sample sizes for some comparisons and unequal variances in each population (see below), the two-tailed Welch's version of the $t$-test is appropriate[50], which compares each sample mean and variance to the $t$ distribution. The non-parametric two-sample KS test compares two samples and measures the maximum distance between their empirical cumulative distributions ($D_{KS}$) and rejects the null hypothesis if $p_{KS}$ exceeds some critical threshold. The Mann–Whitney $U$ test is another non-parametric test that examines two samples (for example, $X$ and $Y$), with the null hypothesis being that a random value from sample $X$ has an equal probability of being greater or less than a random value from population $Y$; this is the expected outcome if the two samples share the same underlying distribution. All three tests compare the hypotheses $H_0$ and $H_1$ and return a $P$ value. If the $P$ value is greater than a defined significance threshold ($\alpha$), typically $\alpha = 0.05$, then the $H_0$ hypothesis cannot be rejected; if the $P$ value is less than the defined significance, then $H_0$ can be rejected at $(1 - \alpha)$ confidence in favour of $H_1$. For our hypotheses, if $P > 0.05$, then it is reasonable to infer that the JH and BGS zircon palaeointensity distributions cannot be distinguished at the 95% confidence threshold.

We first compare the JH and BGS between 3.4 and 3.9 Ga and justify this choice of age range further below; we find that the Welch's $t$-test, non-parametric two-sample KS test and Mann–Whitney $U$ test all yield $P$ values > 0.05, indicating that we cannot reject the null hypothesis $H_0$ (Extended Data Table 2).

In Tarduno et al.[7], JH zircon palaeointensity data were averaged in a moving-window model using 100-Myr non-overlapping bins to estimate the PDM for each 100-Myr interval. Bin edges were defined by the distribution of available data in their set of JH zircon palaeointensities, starting at 3,258 Ma, 3,358 Ma up to 4,258 Ma. We follow this approach to define 100-Myr averaging bins with the BGS data, using identical bin definitions to allow for direct comparison between JH and BGS palaeointensity data. Finally, palaeointensities from both JH and BGS datasets are combined using the same bin definitions to produce 100-Myr averages used in Extended Data Fig. 4. Palaeointensity statistics for each 100-Myr bin are provided in Extended Data Table 3.

We next consider the 100-Myr age bins defined in the moving-window model separately (Fig. 3 and Extended Data Table 2). Only two intervals contain more than five observations (the minimum threshold for Welch's $t$-test) for both the JH and BGS datasets, centred on 3,408 Ma and 3,508 Ma. The Welch's $t$-test for each interval yields $P$ values of 0.17 and 0.09, respectively, and thus again the null hypothesis $H_0$ that the JH and BGS data record the same field intensity cannot be rejected. Non-parametric tests yield similar results, with $P$ values exceeding 0.05 for both intervals (Extended Data Table 2), supporting the interpretation of the Welch's $t$-test.

To define the longest age interval that JH and BGS data sample the same underlying field, a series of Welch's $t$-tests was conducted for data 3.9 Ga and younger. For each test, the age interval was successively shortened by adjusting the younger bound back in time, with the resulting $P$ value recorded (Extended Data Fig. 4a). We find that, for ages older than 3.4 Ga, the $P$ value is high and supports the null hypothesis. We also note that, for age bounds less than 3.4 Ga, the Welch's $t$-test $P$ value drops below 0.05, resulting in the rejection of the null hypothesis $H_0$ in favour of $H_1$ (Extended Data Fig. 4a). This suggests that the JH and BGS data begin to sample different underlying field distributions when the comparison interval extends into younger times.

We next consider whether the JH palaeointensity history predicts that observed from the BGS between 3.9 and 3.4 Ga. We examine the residuals between the 100-Myr smoothing window model for the JH palaeointensity and the data used to define it (that is, the palaeointensity data from the JH zircons; Extended Data Fig. 4b,d). We do the same for the BGS palaeointensity dataset (that is, the BGS palaeointensity data versus the JH model; Extended Data Fig. 4c,e). Given the two distributions, we apply a two-sample KS test (Extended Data Fig. 4f) and assess whether the new data residuals look like the data that were input into the JH smoothed window average.

The null hypothesis is that the two samples (that is, residual populations) are drawn from the same distribution. In the case in which the new data are well predicted by the model, the KS test should return a $P$ value > 0.05 and, if the new data are poorly predicted, the $P$ value is <0.05 (that is, in this case, the hypothesis that the residuals are similar can be rejected). When applied to the new BGS, the $P$ value is 0.32, which supports the hypothesis that the JH model can predict the Barberton zircon results. Although the $P$ value is above the 95% threshold, the distributions nevertheless merit some discussion. The JH palaeointensities are clearly more scattered than the BGS results. This could signal one, or both, of the following: (1) the BGS zircons are, in general, better preserved, having less amorphous iron oxides in cracks than the JH and, therefore, they might be better recorders; (2) the JH might record the field on shorter timescales than the BGS, averaging higher-frequency secular variations of the past geomagnetic field. Palaeointensities from the SCP analyses of Nondweni dacite samples are within the BGS values of the same age, but SCP of Barberton dacites at 3.45 Ga are higher. Both the Nondweni and Barberton SCP values are plotted as individual results from two relatively shallow intrusions, and these might be expected to sample higher-frequency variations of the geomagnetic field than the zircons.

**Plate-motion analysis.** Here we consider the probability of observing two sites, located on different plates, experiencing both little to no latitudinal motion as well as little to no relative latitudinal distance separation during an interval of 600 Myr. To construct this test, the plate-reconstruction model of Merdith et al.[53], a continuous full-plate reconstruction model spanning the past 1 Gyr, is used. A set of sites are defined for the present day and the site palaeolocations are

reconstructed back from the present day to 600 Ma using GPlates[54]. The distribution of palaeolocations is described and comparisons between sites located on different plates is made. From this empirical dataset, a set of statistical tests is constructed.

The sampling grid is defined using a Fibonacci spiral[55], which yields a uniform distribution of locations distributed globally with a median separation of approximately 6°. From this grid, 1,000 sites are assigned to plates using a built-in GPlates function. Sites that do not fall within the boundaries of a plate are removed before the analysis. Using the plate-motion model of Merdith et al.[53], site palaeolocations are reconstructed in 1-Myr steps from the present day to 600 Ma. From the initial set of sites, only those assigned to plates that existed 600 Ma are preserved, resulting in a set of 228 locations (Extended Data Fig. 5a).

We model a plate-motion path for each site from the GPlates reconstruction. Using the plate-motion path, the absolute maximum latitudinal distance travelled, $\Delta\lambda$ can be determined as:

$$\Delta\lambda = \max(|\lambda_0 - \lambda_{P_t}|) \tag{3}$$

in which $\lambda_0$ is the present-day site latitude, $\lambda_{P_t}$ is the palaeolatitude at time $t$ and 'max' is the maximum distance for the set considered. $\Delta\lambda$ distances range from 33° to 127°, with a median $\Delta\lambda$ distance of 76° (Extended Data Fig. 6b).

From the distribution of $\Delta\lambda$ for each of the 228 sites located on 66 unique 'plates' as designated by GPlates, the general trend of latitudinal motion can be described (Extended Data Fig. 5c). Broadly speaking, there is a weak positive correlation between the magnitude of $\Delta\lambda$ and the age at which the maximum latitudinal distance is observed (Pearson's correlation coefficient, $r = 0.49$);, however, sites with $\Delta\lambda$ falling in the lower 5% of the distribution ($\Delta\lambda \le 40°$, $n = 12/228$) have ages ranging from 150 to 480 Ma. We note that $\Delta\lambda$ estimates can be biased downward by as much as approximately 10° when sites are downsampled at 100-Myr versus 1-Myr intervals.

With these general trends in mind, we next consider whether any sites experience little to no latitudinal motion. No sites could be identified with near-zero latitudinal motion irrespective of the sampling level (Extended Data Fig. 6a,b). If we arbitrarily define 'little motion' as ≤30° of latitudinal motion, no sites at 1-Myr or 20-Myr sampling meet the criterion, whereas only a single site (located on the Paraná/Pampia plate) met the criterion with 100-Myr downsampling. If the definition of 'little' motion is further relaxed to a threshold of ≤35° and the sampling interval to 100 Myr, 11 of 228 sites meet the criterion. The 11 sites showing ≤35° motion are observed on seven plates. We note that we only consider the maximum amount of latitudinal motion in this analysis; if both northern and southern latitudinal motion are considered separately instead, the lower 5th percentile of the revised distribution indicates a minimum total latitudinal motion of approximately 50–60°.

Given the rarity, or absence, of plates showing no motion over 600 Myr, the likelihood of observing two sites on separate plates with little to no latitudinal motion is very low. We next consider the bounds of $\Delta\lambda$ and relative latitudinal distance for plate pairs provided by the palaeointensity data (Fig. 4). The case of a plate backtracking to the north is automatically recognized as being inconsistent with plate motion over the past 600 Myr by the single-site $\Delta\lambda$ distributions (Extended Data Fig. 6a,b). For a plate backtracking to the south, the $\Delta\lambda$ bound is 48° (see main text). The bound on relative distance between the plate pair is the combination of this backtracked southward motion for one plate and the maximum backtracked northward motion of another plate consistent with the palaeointensity data (5°), for a total of 53°.

To determine the probability of sampling sites from two separate plates with these $\Delta\lambda$ and relative distance characteristics, a hypergeometric distribution is appropriate. A hypergeometric distribution models a sampling without replacement, in contrast to the more common binomial distribution, which samples trials with replacement[56]. Sampling without replacement is appropriate for this scenario because the same site (plate) cannot be selected twice. The probability mass function ($p(k)$) for a hypergeometric function is defined as:

$$p(k) = \frac{\binom{K}{k}\binom{N-K}{n-k}}{\binom{N}{n}} \tag{4}$$

in which $N = 66$ is the number of plates, $K$ is the number of plates in the 600-Myr dataset known to record latitudinal motion and relative distance less than or equal to a specified threshold, $n = 2$ represents the number of unique plates selected in a single random sampling and $k$ is the number of plates in that sampling showing limited latitudinal motion. The resulting probability of identifying a single pair of plates ($k = 2$) that both show limited latitudinal motion, here defined as $\Delta\lambda \le 48°$ and relative latitudinal distance ≤53° at 100-Myr downsampling, is <1% (Fig. 4f).

Finally, we note that palaeolongitudinal trends characteristic of plate tectonics can also be inferred from plate-motion models spanning the past 600 Myr. To avoid biasing of longitudinal motion owing to geometric effects (that is, less area at higher latitudes), we normalize longitudinal motion by analysing the fraction of maximum angular travel from the initial position that is not explained by latitudinal change. We note that plates having dominant longitudinal motion (for example, >50%) are rare (about 5%). In these cases, there is at least 40° of latitudinal motion.

**Prior geologic models and attendant implications.** A 'Vaalbara' supercraton involving the Pilbara and Kaapvaal cratons between about 3.1 and 2.7 Ga has been proposed by some authors[57,58] but challenged by others[59]. Arguably the strongest geologic correlations are between the approximately 2.7–2.8-Ga volcanism on these cratons (Ventersdorp Supergroup of the Kaapvaal Craton and Fortescue Group of the Pilbara Craton). Palaeomagnetic data for the Kaapvaal Craton, applying a tilting interpretation[59], imply emplacement at a very high latitude (75.4°). There is an approximately 700-Myr-long age gap between this volcanic association and the time when the BGS and JH zircon records start to diverge, which might hint at a start of horizontal motions characteristic of plate tectonics. This duration is 100 Myr longer than that over which we used to assess the latitudinal-motion characteristics of plate tectonics. Nevertheless, our analysis can be used as a conservative measure to assess whether plate-motion rates typical of plate tectonics are compatible with a high-latitude Kaapvaal Craton in the Neoarchaean and a low-latitude position in the Palaeoarchaean. Given that the median maximum latitudinal displacement from 0–600 Ma is 76°, it is clear that plate-motion rates are compatible. Assuming a location of the Kaapvaal craton solely in one (N or S) hemisphere yields an average latitudinal component of motion of 8 mm year⁻¹. Assuming one equatorial crossing results in an average latitudinal component of motion of 16 mm year⁻¹.

**Palaeointensity variations and dynamo efficiency.** Palaeomagnetic variations, principally the occurrence of reversing and seemingly non-reversing periods (superchrons), but also inferences on secular variation, have long been used to argue for changes in core–mantle boundary structure influencing the efficiency of the geodynamo[60,61]. Palaeointensity variations of superchron versus reversing intervals[62–65] have also been interpreted as reflecting these variations, and this is supported by some numerical geodynamo models[66]. More recently, it has been suggested that such changes may extend to the Devonian[67] and/or might have started in the Ediacaran Period[19]. If palaeointensity variations similar to those observed in the Phanerozoic had been present in our 3.9–3.4-Ga data, we might not have been able to separate changes in dynamo efficiency from palaeolatitude changes. But because the palaeointensity record is constant, we infer that palaeolatitudes

were constant and mantle processes were not creating the latitudinal core–mantle boundary heat-flux patterns necessary to greatly affect geodynamo efficiency.

## Data availability

Data presented here are available in the Earthref (MagIC) database (earthref.org/MagIC/19526; https://doi.org/10.7288/V4/MAGIC/19526). Source data are provided with this paper.

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

**Acknowledgements** We thank G. Kloc for the sample preparation, T. Pestaj in the SHRIMP laboratory and M. Polivchuk in the GSC field emission SEM laboratory. This work was performed in part at the University of Rochester Integrated Nanosystems Center (URnano), which is supported by the University of Rochester. A.H. thanks the Mpumalanga Tourism and Parks Agency and Sappi Forests for access and support. R.K.B. acknowledges support from the Leverhulme Early Career Fellowship (ECF-2020-617). This work was supported by the National Science Foundation (grant nos. EAR-1656348 and EAR-2051550 to J.A.T.). This is NRCan contribution number 20220385.

**Author contributions** J.A.T. conceived and supervised the overall study. J.A.T., A.H. and J.J. conducted the field studies. M.I.-M. and R.D.C. separated zircons. R.D.C. conducted palaeomagnetic measurements and these data were analysed by R.D.C. and J.A.T. Reflected-light studies were conducted by R.D.C. and electron-microscope analyses by T.Z., J.A.T. and R.D.C. H.O. assisted in methods. Age data were collected by N.R. and analysed by N.R., W.J.D. and J.A.T. R.K.B. conducted statistical and plate-motion analyses. F.N. assisted in geodynamic interpretations, M.K.W. in tectonic implications and G.M. in microtectonic analysis. J.A.T. wrote the manuscript, with contributions from all the authors.

**Competing interests** The authors declare no competing interests.

**Additional information**
**Correspondence and requests for materials** should be addressed to John A. Tarduno.

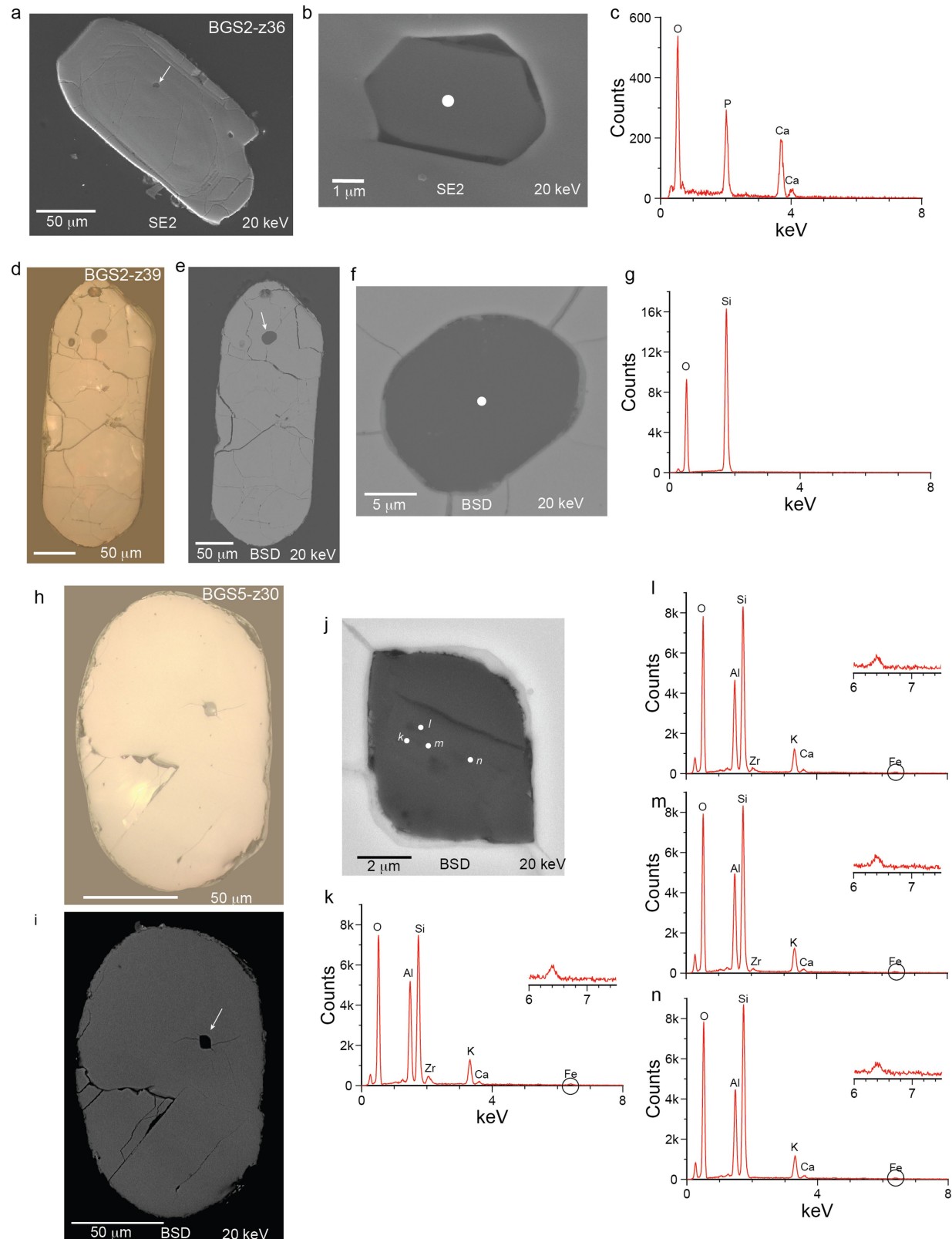

**Extended Data Fig. 1 | Silicate and apatite inclusions in BGS detrital zircons.** **a**, SEM secondary electron (SE2) images of zircon BGS2-z36 with apatite inclusion highlighted. **b**, Higher magnification of the inclusion with EDS analysis location highlighted. **c**, EDS spectra. **d**, Reflected-light microscope image of zircon BGS2-z39. **e**, SEM BSD image of zircon BGS2-z39 with quartz inclusion highlighted. **f**, Higher magnification of the inclusion with EDS analysis location highlighted. **g**, EDS spectra. **h**, Reflected-light microscope image of zircon BGS5-z30. **i**, SEM BSD image of zircon BGS5-z30 with feldspar inclusion identified. **j**, Higher magnification of the inclusion with EDS analysis locations highlighted. **k**–**n**, EDS spectra.

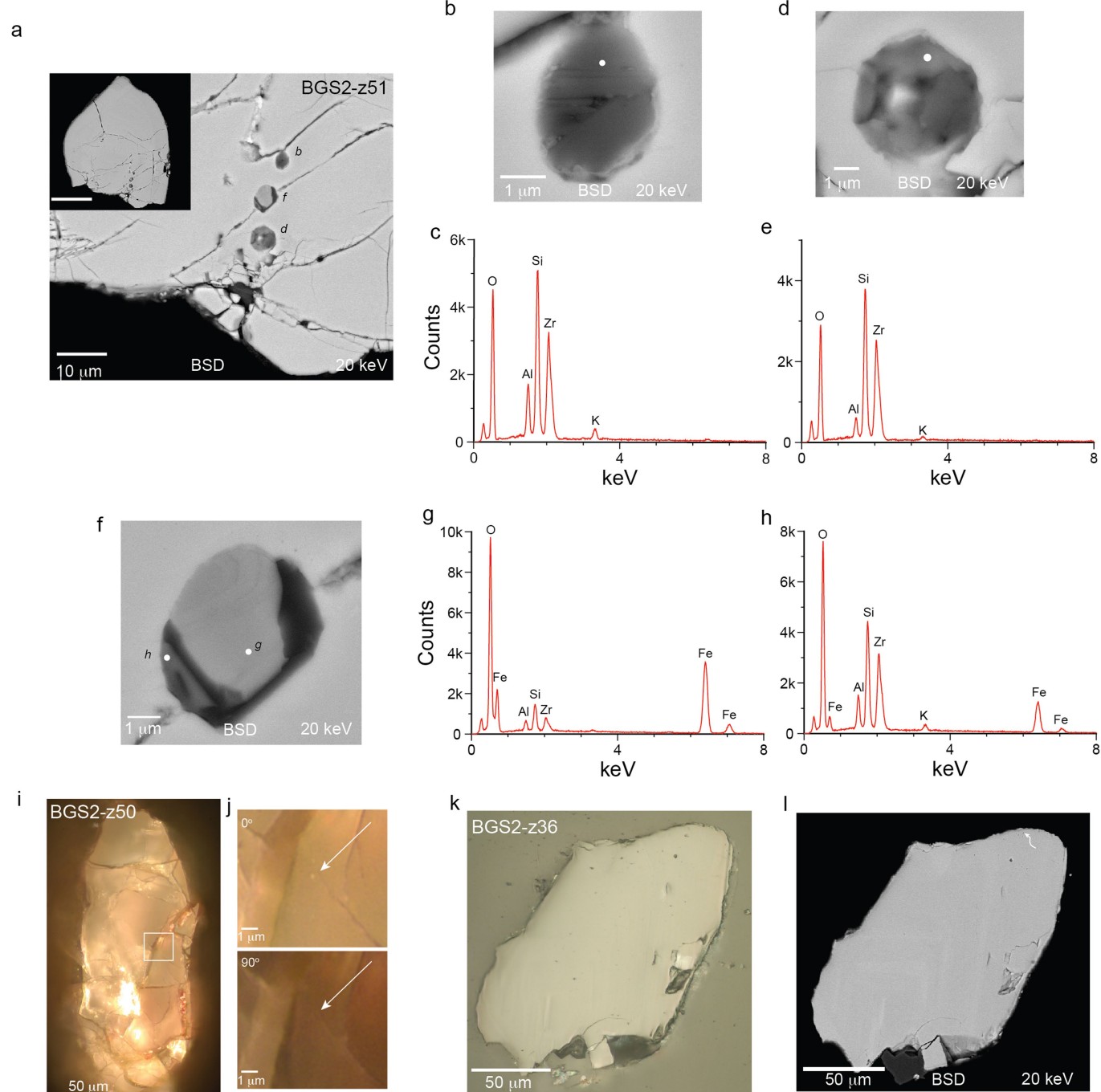

**Extended Data Fig. 2 | Fe-oxide and silicate inclusions in BGS detrital zircons. a**, SEM BSD image of zircon BGS2-z51 highlighting melt inclusions discussed (**b**,**f**,**d**); entire grain shown as inset with 50-µm scale bar. **b**, Higher-magnification SEM BSD image of feldspar inclusion b. EDS analysis location highlighted. **c**, EDS spectra of inclusion b. **d**, Higher-magnification SEM BSD image of multicomponent inclusion partially disrupted by polishing. EDS analysis location highlighted. **e**, EDS spectra from **d** suggesting a feldspar composition. **f**, Higher-magnification SEM BSD image of inclusion f (see Fig. 1 showing extinction with 90° change in polarization) showing EDS analysis

locations. This inclusion may be partially disrupted by polishing. **g**, EDS spectra of main grain in **f** highlighting strong Fe signal. Al may be from adjacent feldspar (see **h**). **h**, EDS of small grain in **f** showing spectra compatible with feldspar. **i**, Reflected-light microscopy image (100×) of zircon BGS2-z50. **j**, Reflected-light images (1,000× oil immersion) at 0° (top) and 90° (bottom) polarization showing extinction of Fe-oxide inclusion (magnetite) at depth (see Fig. 1). **k**, Reflected-light image of zircon BGS5-z36 (see Fig. 1). **l**, SEM BSD image of BGS5-z36. Fe particles shown in Fig. 1 are approximately 2 µm from the grain edge (also see 'Microtectonic analyses of zircons' section in Methods).

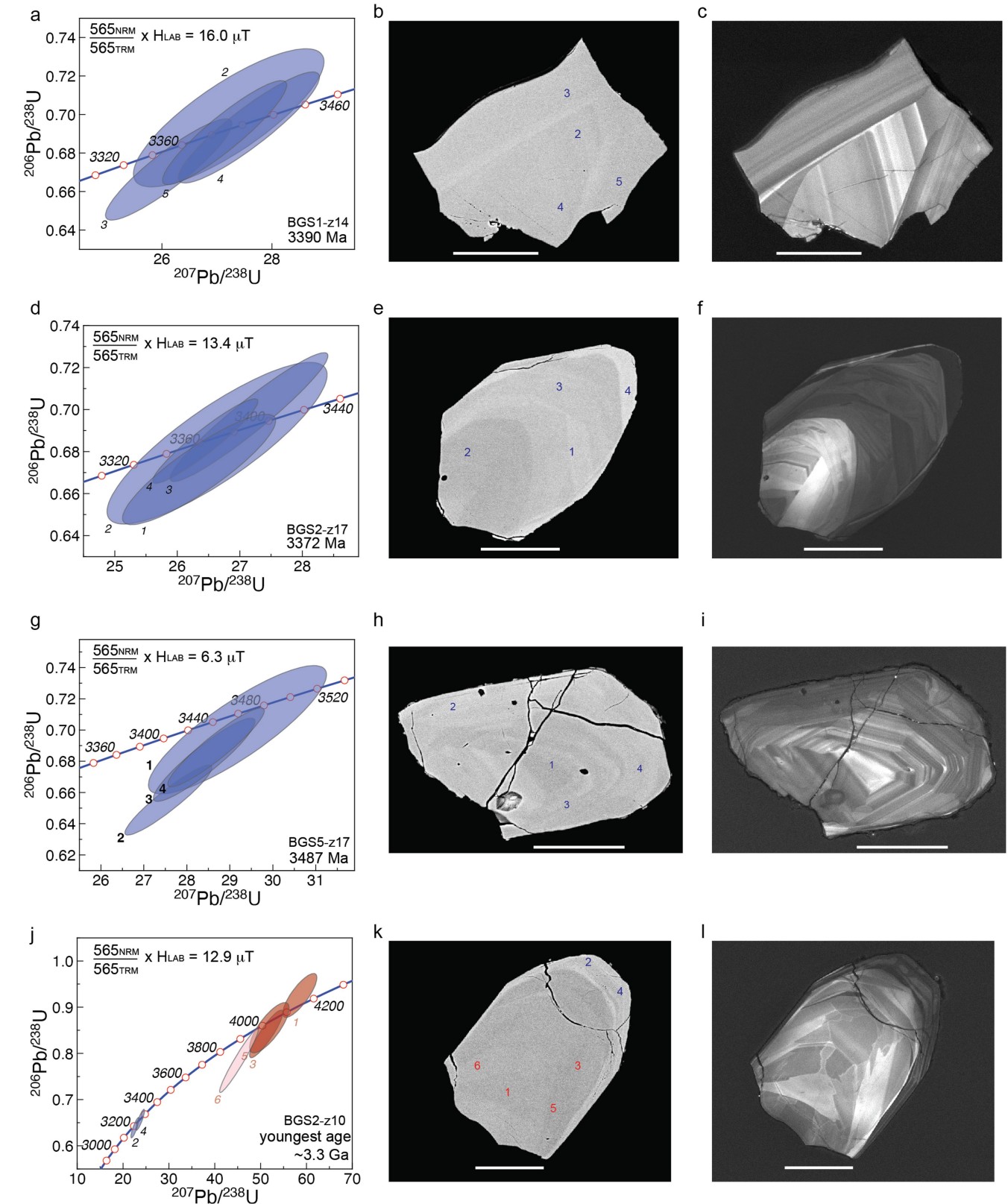

**Extended Data Fig. 3 | Further palaeointensity determinations and SHRIMP age data from individual BGS zircon crystals. a**, Concordia diagram showing SHRIMP geochronological analyses (uncertainty ellipses are 2σ) and palaeointensity value for zircon BGS1-z14. **b**, Corresponding backscatter scanning electron microscope with analysis spots labelled in **a**. **c**, Corresponding cathodoluminescence image. Images **b** and **c** shown with 50-μm scale. **d–f**, Analyses as shown in **a–c** for zircon BGS2-z17. **g–i**, Analyses as shown in **a–c** for zircon BGS5-z17. **j–l**, Analyses as shown in **a–c** for zircon BGS2-z10. Red, analyses from core; blue, analyses from rim.

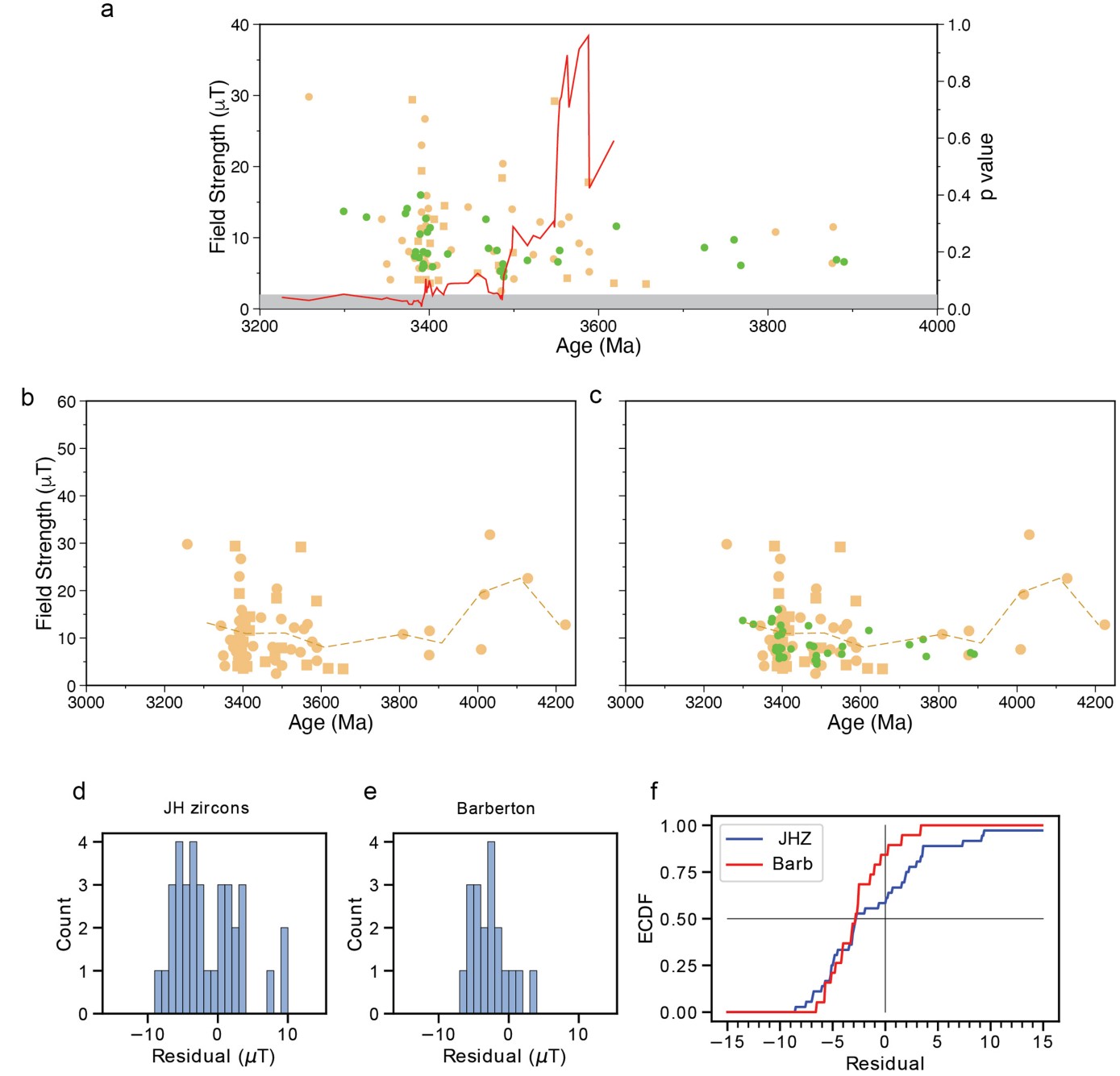

**Extended Data Fig. 4 | BGS and JH zircon palaeointensity values versus time. a**, BGS palaeointensity data (green circles) and JH palaeointensity data (yellow circles), shown with Welch's *t*-test *P*-value (red curve; see 'Statistical analysis of BGS and JH zircon palaeointensity data' section in Methods). **b**, JH zircon palaeointensity values (Tarduno et al.[6,7]) with 100-Myr moving-window average. **c**, BGS zircon palaeointensity values shown with data of **a**. **d**, Residuals of JH palaeointensity values relative to model (JH 100-Myr moving-window average). **e**, Residuals of BGS palaeointensity values relative to model (JH 100-Myr moving-window average). **f**, Empirical cumulative distribution function (ECDF) plot of JH and BGS residuals.

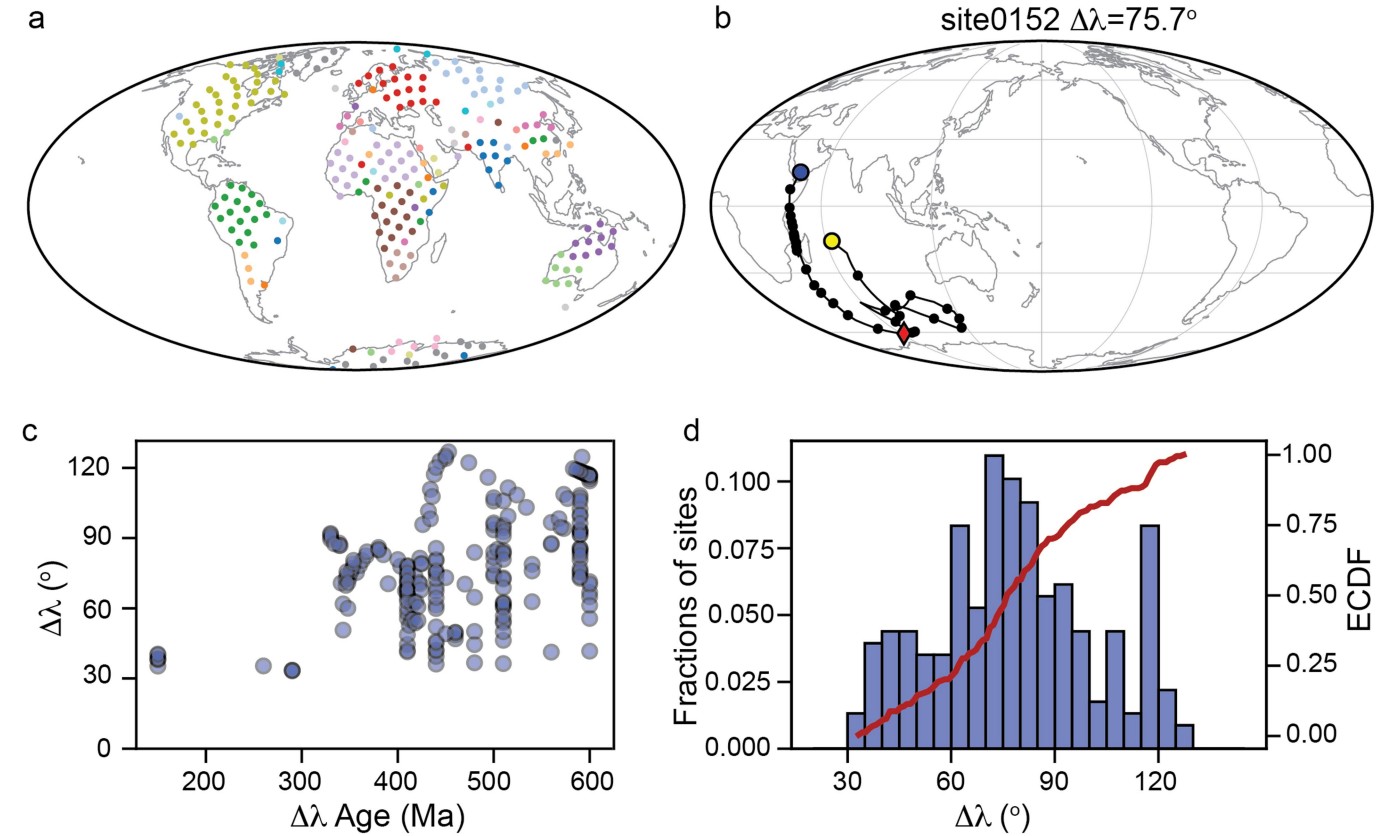

**Extended Data Fig. 5 | Plate-motion analysis 0–600 Ma. a**, Distribution of sites used in plate-motion analysis. Different colours distinguish between assigned plates. **b**, Representative example of the motion path for a single site with $\Delta\lambda$ at the median value (76°) determined using the plate-motion models described in the text. Solid line shows motion path resolved at 1-Myr intervals, black circles show 20-Myr steps. Blue circle, present-day site location; yellow circle, site palaeolocation at 600 Ma; red diamond, location at which the maximum latitudinal displacement is reached. **c**, Distribution of $\Delta\lambda$ from 228 sites located on 66 plates, determined using 1-Myr time steps shown with age of maximum latitudinal distance ($\Delta\lambda$) from present versus angle. **d**, Distribution of $\Delta\lambda$ in 5° bins; red line (and right $y$ axis) shows empirical cumulative distribution of $\Delta\lambda$.

a
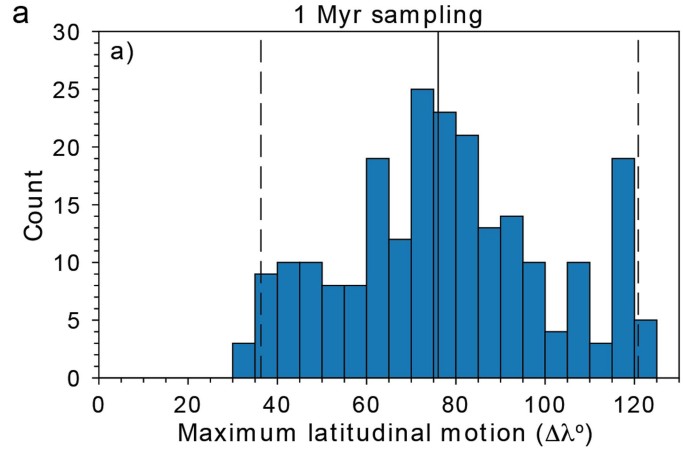

b
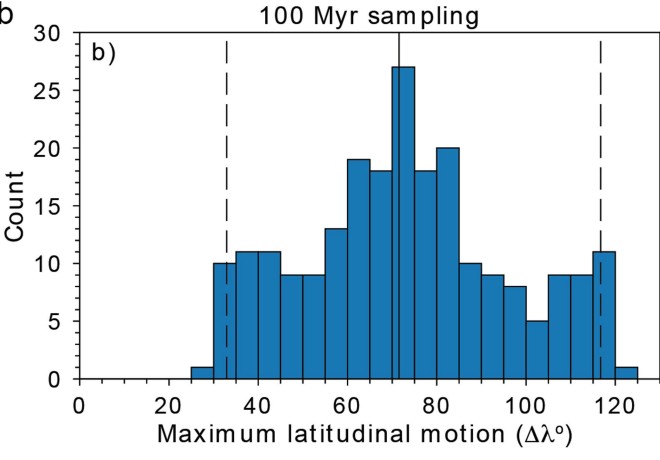

**Extended Data Fig. 6 | Maximum latitudinal motion (Δλ) of sites for the past 600 Myr.** Plate-motion-model-derived Δλ values with (**a**) 1-Myr and (**b**) 100-Myr downsampling. Histograms show number of sites (total *n* = 228) with Δλ binned into 5° groups. Solid vertical line shows median Δλ; dashed vertical lines mark the 95% interval for the distribution.

## Extended Data Table 1 | Palaeointensity determinations

| Sample | Age | Pint | NRM | 565$_{off}$ | | | 565$_{on}$ | | | $\delta_{565}$ | MD$_{ck}$ |
|---|---|---|---|---|---|---|---|---|---|---|---|
| | Ma | $\mu T$ | A m$^2$ | $x_{ave}$ | $y_{ave}$ | $z_{ave}$ | $x_{ave}$ | $y_{ave}$ | $z_{ave}$ | o | % |
| BGS1-z3 | 3398 | 7.8 | 2.218E-12 | -6.387E-13 | -6.125E-13 | -5.212E-13 | -7.972E-13 | -1.364E-12 | -2.351E-12 | 22.8 | 1.2 |
| BGS1-z4 | 3396 | 12.7 | 2.329E-12 | 4.694E-13 | -7.874E-13 | 6.348E-13 | 7.221E-13 | -8.545E-13 | 1.922E-12 | 11.5 | 1.3 |
| BGS1-z7 | 3383 | 7.4 | 1.238E-12 | -9.994E-14 | 2.936E-13 | -5.482E-13 | -2.385E-13 | 6.380E-13 | -1.765E-12 | 17.0 | 1.5 |
| BGS1-z9 | 3881 | 6.9 | 1.083E-12 | 4.364E-13 | 5.238E-13 | -2.727E-13 | 5.526E-13 | 8.101E-13 | -1.837E-12 | 11.2 | 2.3 |
| BGS1-z11 | 3768 | 6.1 | 1.381E-12 | -2.365E-13 | -6.428E-13 | -4.849E-13 | -1.582E-13 | -1.046E-12 | -2.517E-12 | 11.4 | 1.6 |
| BGS1-z12 | 3404 | 5.9 | 1.523E-12 | -2.786E-13 | -7.782E-15 | -5.896E-13 | -5.598E-13 | -1.759E-14 | -2.230E-12 | 9.7 | 1.9 |
| BGS1-z14 | 3390 | 16.0 | 1.832E-12 | 5.771E-13 | -7.334E-13 | 8.949E-13 | 5.744E-13 | -6.632E-13 | 2.108E-12 | 3.3 | 2.1 |
| BGS1-z15 | 3393 | 8.0 | 1.138E-12 | 2.889E-13 | -6.339E-13 | -2.183E-13 | 2.676E-13 | -2.695E-13 | -1.546E-12 | 15.4 | 0.9 |
| BGS2-z1 | 3422 | 7.7 | 1.823E-12 | 1.316E-13 | -5.088E-13 | -7.562E-13 | 2.983E-13 | -6.516E-13 | -2.533E-12 | 7.0 | 1.3 |
| BGS2-z2 | 3621 | 11.6 | 2.528E-12 | 3.269E-13 | 9.080E-13 | -5.024E-13 | 4.935E-13 | 1.209E-12 | -1.872E-12 | 14.1 | 1.3 |
| BGS2-z3 | 3467 | 12.6 | 2.034E-12 | -7.318E-13 | -4.362E-13 | -7.225E-13 | -6.741E-13 | -4.862E-13 | -2.047E-12 | 3.3 | 2.9 |
| BGS2-z4 | 3388 | 7.2 | 2.183E-12 | 3.844E-13 | 6.713E-13 | -5.397E-13 | 2.498E-13 | 4.601E-13 | -2.484E-12 | 7.3 | 3.0 |
| BGS2-z5 | 3393 | 6.3 | 1.782E-12 | -1.222E-13 | 4.317E-13 | 7.833E-13 | -1.623E-13 | 2.713E-13 | 2.910E-12 | 4.4 | 0.4 |
| BGS2-z7 | 3470 | 8.5 | 1.429E-12 | 1.455E-13 | 1.229E-13 | -5.291E-13 | 1.583E-13 | 1.692E-13 | -1.514E-12 | 2.8 | 0.3 |
| BGS2-z8 | 3488 | 4.5 | 1.183E-12 | -6.521E-13 | -2.464E-13 | 2.305E-13 | -6.659E-13 | -4.802E-13 | 2.669E-12 | 5.5 | 2.0 |
| BGS2-z10 | 3226 | 12.9 | 1.492E-12 | 2.625E-13 | -4.855E-13 | -2.423E-13 | 3.362E-13 | -4.661E-13 | -9.415E-13 | 6.2 | 2.9 |
| BGS2-z11 | 3552 | 6.6 | 1.194E-12 | 7.634E-14 | 4.466E-13 | -3.829E-13 | 2.203E-13 | 8.706E-13 | -1.654E-12 | 19.4 | 0.7 |
| BGS2-z13 | 3384 | 8.0 | 1.004E-12 | -4.144E-13 | 2.923E-13 | -6.086E-13 | -6.833E-13 | 2.915E-13 | -2.063E-12 | 10.5 | 0.7 |
| BGS2-z14 | 3890 | 6.6 | 1.103E-12 | 2.487E-13 | -5.237E-13 | 1.630E-13 | 3.019E-13 | -5.065E-13 | 1.528E-12 | 2.3 | 2.2 |
| BGS2-z15 | 3760 | 9.7 | 1.236E-12 | 4.188E-13 | -5.098E-13 | -3.347E-13 | 8.339E-13 | -8.081E-13 | -1.360E-12 | 26.5 | 1.8 |
| BGS2-z17 | 3372 | 13.4 | 1.073E-12 | -2.099E-13 | 5.614E-13 | -3.488E-13 | -2.004E-13 | 4.338E-13 | -1.115E-12 | 9.5 | 2.7 |
| BGS2-z19 | 3374 | 14.1 | 1.033E-12 | -2.155E-13 | 2.095E-13 | 5.227E-13 | -2.886E-13 | 2.448E-13 | 1.158E-12 | 7.3 | 0.6 |
| BGS2-z21 | 3480 | 8.2 | 1.300E-12 | -4.562E-13 | -1.688E-13 | -5.479E-13 | -5.556E-13 | -4.436E-13 | -1.862E-12 | 12.5 | 3.7 |
| BGS2-z22 | 3394 | 6.0 | 9.394E-13 | 3.909E-13 | -2.647E-13 | 4.221E-13 | 7.136E-13 | -1.819E-13 | 1.959E-12 | 12.2 | 0.9 |
| BGS5-z1 | 3554 | 8.2 | 2.419E-12 | 3.169E-13 | 8.213E-13 | 3.846E-13 | 4.005E-14 | 9.555E-13 | 2.118E-12 | 10.1 | 2.4 |
| BGS5-z5 | 3398 | 10.8 | 2.028E-12 | -2.486E-13 | 9.754E-13 | 7.207E-13 | -4.905E-13 | 8.996E-13 | 2.426E-12 | 8.5 | 1.0 |
| BGS5-z6 | 3516 | 6.8 | 2.422E-12 | -5.056E-13 | 5.021E-13 | -6.834E-13 | -5.189E-13 | 7.090E-13 | -2.856E-12 | 5.5 | 0.7 |
| BGS5-z7 | 3299 | 13.7 | 2.092E-12 | 7.491E-14 | -9.963E-13 | -9.548E-13 | 1.893E-13 | -9.831E-13 | -2.466E-12 | 4.4 | 1.3 |
| BGS5-z8 | 3401 | 11.4 | 1.982E-12 | -1.633E-13 | 3.703E-13 | -9.221E-13 | -3.887E-13 | 6.573E-13 | -2.193E-12 | 16.0 | 7.2 |
| BGS5-z12 | 3725 | 8.6 | 1.325E-12 | -1.959E-13 | 5.268E-13 | -4.801E-13 | -6.090E-13 | 5.542E-14 | -1.610E-12 | 29.0 | 1.2 |
| BGS5-z14 | 3488 | 5.3 | 1.327E-12 | -3.558E-13 | 3.049E-13 | 2.760E-13 | -8.782E-13 | 7.214E-13 | 1.672E-12 | 25.6 | 15.2 |
| BGS5-z15 | 3484 | 5.3 | 1.102E-12 | 2.141E-13 | 3.738E-13 | -4.175E-13 | 5.542E-13 | 1.029E-12 | -1.938E-12 | 25.9 | 1.7 |
| BGS5-z17 | 3487 | 6.3 | 1.437E-12 | -4.009E-13 | -4.898E-13 | 5.425E-13 | -5.928E-13 | -1.056E-12 | 2.451E-12 | 17.4 | 0.5 |
| BGS5-z18 | 3392 | 5.7 | 1.061E-12 | 2.312E-13 | 3.454E-13 | 3.259E-13 | 7.688E-15 | 6.292E-13 | 1.657E-12 | 15.2 | 3.9 |
| BGS5-z19 | 3389 | 10.5 | 9.382E-13 | -1.955E-13 | 3.787E-13 | -4.603E-13 | -1.892E-13 | 3.914E-13 | -1.354E-12 | 0.9 | 2.7 |

Three-axis magnetometer data for NRM intensity, field off (565$_{off}$) and field on (565$_{on}$) steps, averaged for several measurements ($x_{ave}$, $y_{ave}$, $z_{ave}$). Magnetometer values are listed in electromagnetic units (that is, raw magnetometer data). $\delta_{565}$ represents the calculated angle between the TRM vector (565$_{on}$-565$_{off}$) and the applied field. MD$_{ck}$ represents the % difference of a second 565$_{off}$ step to check for multidomain affects.

**Extended Data Table 2 | BGS/JH comparisons: statistical parameters**

| Young | Old | $N_{JH}$ | $N_{BGS}$ | $D_{WT}$ | $p_{WT}$ | $D_{KS}$ | $p_{KS}$ | $D_{MWU}$ | $p_{MWU}$ |
|-------|-----|-------|--------|-------|-------|-------|-------|---------|---------|
| 3358 | 3458 | 33 | 16 | 1.382 | 0.174 | 0.231 | 0.525 | 297.000 | 0.488 |
| 3458 | 3558 | 14 | 10 | 1.821 | 0.087 | 0.329 | 0.467 | 87.000 | 0.333 |
| 3400 | 3900 | 36 | 19 | 1.674 | 0.100 | 0.256 | 0.322 | 376.000 | 0.553 |

Young, youngest palaeointensity sample in interval (Myr); Old, oldest palaeointensity sample in interval (Myr); $N_{JH}$, $N_{BGS}$, number of samples included in interval for JH and BGS data, respectively; $D_{WT}$, $p_{WT}$, Welch's Student $t$-test statistic, $P$-value; $D_{KS}$, $p_{KS}$: two-sample KS test statistic, $P$-value; $D_{MWU}$, $p_{MWU}$, Mann–Whitney $U$ test statistic, $P$-value.

**Extended Data Table 3 | 100-Myr averages of JH, BGS and JH+BGS data**

| JH | | | | BGS | | | | JH+BGS | | | |
|---|---|---|---|---|---|---|---|---|---|---|---|
| Bin midpoint (Ma) | N obs. | Mean Field ($\mu$T) | SEmean ($\mu$T) | Bin midpoint (Ma) | N obs. | Mean Field ($\mu$T) | SEmean ($\mu$T) | Bin midpoint (Ma) | N obs. | Mean Field ($\mu$T) | SEmean ($\mu$T) |
| 3208 | - | - | - | 3208 | 1 | 12.9 | - | 3208 | 1 | 12.9 | - |
| 3308 | 4 | 13.2 | 5.0 | 3308 | 1 | 13.7 | - | 3308 | 5 | 13.3 | 4.0 |
| 3408 | 33 | 11.0 | 1.1 | 3408 | 16 | 9.1 | 0.8 | 3408 | 49 | 10.3 | 0.8 |
| 3508 | 14 | 11.1 | 1.9 | 3508 | 10 | 7.2 | 0.7 | 3508 | 24 | 9.5 | 1.2 |
| 3608 | 8 | 8.1 | 1.7 | 3608 | 1 | 11.6 | - | 3608 | 9 | 8.5 | 1.5 |
| 3708 | - | - | - | 3708 | 1 | 8.6 | - | 3708 | 1 | 8.6 | - |
| 3808 | 1 | 10.8 | - | 3808 | 2 | 7.9 | 1.3 | 3808 | 3 | 8.9 | 1.2 |
| 3908 | 2 | 9.0 | 1.8 | 3908 | 2 | 6.8 | 0.1 | 3908 | 4 | 7.9 | 1.1 |
| 4008 | 3 | 19.5 | 5.7 | | | | | 4008 | 3 | 19.5 | 5.7 |
| 4108 | 1 | 22.6 | - | | | | | 4108 | 1 | 22.6 | - |
| 4208 | 1 | 12.8 | - | | | | | 4208 | 1 | 12.8 | - |

.