## [Peer Review File · Nature]

Manuscript Title: Hadean to Palaeoarchean stagnant lid tectonics revealed by zircon magnetism

Reviewer Comments & Author Rebuttals

Reviewer Reports on the Initial Version:

Referee #1:

I can follow the line of arguments developed in the study. The interpretation is reasonable and might stand the test of time. The 'plate tectonic test' (to make it highly unlikely that plate motion would have occurred during 3.9-3.4 Ga) is performed with Phanerozoic plate motion models. This is logical since plate tectonics for sure was active during the entire Phanerozoic. However, the BGS and JH sample locations are associated with the Kaapvaal and Jilgarn-Pilbara cratons for which the oldest APWP has been proposed for a period between ~3.1 and ~2.7 Ga: they would be juxtaposed and constitute the Vaalbara super-craton (e.g. Zegers et al., 1998; de Kock et al., 2009). OK, this is younger than the time span considered in the present study and the APWP is not beyond discussion (e.g. Evans and Muxworthy, 2019).

Some discussion on how the paleolatitudes compare and what would be the paleolatitude trajectory of the Vaalbara super-craton between ~3.1 and ~2.7 Ga would add. Plate motion of several cm per year to over 1 m per year have been argued for (the latter is very high for present-day standards). Is the change over from stagnant-lid tectonics to (present-day-style?) plate tectonics occurring within a few hundreds of millions of years? What if present-day NW Australia and South Africa were juxtaposed to each other already before ~3.1 Ga? OK, they would be stationary (latitudinally) based on the paleointensity results presented here, but is it then warranted to infer a global signal? What are thoughts on when plate tectonics (implicitly present-day style?) would have initiated?

Below are specific comments.

First paragraph

With the onset of plate tectonics present-day style plate tectonics is implied? In the stagnant-lid regime there was volcanism as well? So, nutrient supply (not per se recycling) was possible?

Main text

P2 first paragraph. I can follow very well the line of reasoning developed but there is quite some generic paleomagnetic knowledge assumed implicitly. That the field is dipolar and that inclination and intensity thus scale with latitude. This probably might be presumed knowledge. Does the same also pertain to that individual detrital grains in a sediment cannot provide directional information on paleolatitude (probably clear) but that paleointensity still can do? This might warrant a concise explanation.

References 16 and 17 refer to much younger rocks than dealt with here?

What is implied with 'higher grades'? The younger sedimentary units have not been or barely metamorphosed?

Second paragraph. Zircon by itself cannot provide paleomagnetic information, it is the magnetite inclusions that are responsible for that.

Third paragraph. That the inclusions have not been magnetically reset is crucial. Perhaps it is wise to briefly recapitulate why the grains are not reset. Playing devil's advocate: the reason for the non-varying paleointensity between 3.9 and 3.4 Ga is that the crystal radiometric ages vary between those two values, but the magnetic age would be 3.4 Ga for all of them.

P3 top paragraph. The isolated iron particles are not consisting of metallic iron isn't it? Be specific as to their magnetite nature I would say.

Second paragraph and throughout text: recalculate emu to Am^2 since that is the proper SI unit.

The reason to adopt the 565°C paleointensity protocol in the Jack Hills studies is to see through later medium grade metamorphism. Metamorphic temperatures that prevailed require laboratory heating to 565°C for demagnetization and remagnetization experiments to obtain meaningful results. What is the discriminative power of the paleointensity at that temperature? The 565-580°C portion of the blocking/unblocking temperature spectrum is small. In the Barberton Greenstone Belt lower metamorphic temperatures prevailed so that in principle a larger portion of the demagnetization spectrum can be used for meaningful partial thermoremanent magnetization acquisition experiments: how low a laboratory temperature is required to see through the metamorphism in the Barberton Greenstone Belt? Do the paleointensity results at let's say 520°C deliver the same outcome as those at 565°C?

Third paragraph: the zircons with a much younger rim (how much is much?) presumably do not qualify. OK, the paleointensity is representative of the youngest age.

P4 discussion.

Perhaps it is wise to provide some more information on plate tectonics vs stagnant lid tectonics. Are these seen as the only two options? What is the difference between a mantle plume and a heat pipe?

P5 top. But just before the constant paleointensity has been used to argue for no (significant) plate motion and here it is related to geodynamo efficiency. How can somebody tell apart the difference? The reasoning in the final bits may be a little quickly developed.

P5 sample preparation and selection. What is the reason for picking a NRM cut-off in the first place? Just practical issues? If the samples are not sufficiently magnetized they will deliver meaningless results, or is there yet another reason?

P6 geochronology. Curiosity, where does the geochronological SQUID3 acronym stand for?

Figure 1 caption. Fe is not metallic Fe, name it magnetite or Fe-oxide?

Figure 2. In case of zoned zircons, the youngest growth rim has magnetically reset all older portions of the respective grain? Zircon grows at higher temperatures than the maximum magnetite (un)blocking temperature? What are the demagnetization diagrams of the dated grains shown in c), f) and i)? In a) and b) (added vector difference sum decay curve?) the 565°C value is almost 0.5 of the starting NRM, what is that for c), f) and i)? Show the age dating of BGZ5-z1?

Figure 3. Why are the other zircon paleointensity data shown in grey? Are they from yet another region? They seem to be 'high-ish'. How are the uncertainty ranges in the paleointensity calculated? 1 sigma, 2 sigma? It seems that the means of binned intervals do not come with an uncertainty, yet in extended data table 4 most bins contain more than a single value.

Extended data figures 1-3. Put the respective zircon grain labels on the appropriate panels.

P12. Extended data figure 2. It is not metallic Fe inclusions.

Extended data figure 4. Here paleointensity data are plotted without uncertainty intervals.

Extended data figure 5. What is the red diamond in panel b)?

Extended data table 2. Delta565 is the angle between the TRM vector and the applied field. It shows values up to 29°. Ideally it should be close to zero. What is the underlying reason for the measured deviations? The amount of MD effects seems very low.

Supporting information.

P3. Third paragraph. What is the meaning of acronym PDM? Check other acronyms throughout the document.

P5. Third paragraph. Presumably the likelihood of observing two sites with no motion on separate plates is low. Is that written as such?

Referee #2:

Summary. The manuscript submitted by Tarduno et al. (2022-05-07225; Hadean to Paleoproterozoic stagnant lid tectonics recorded by the paleomagnetism of single zircons of South Africa and Australia) aims to use paleointensity data from Hadean- to Mesoarchean-age single zircons to infer that plate/mobile lid tectonics (as we know it) was not operating at this time. Instead, they argue that stagnant lid tectonics was operational. To do this, they obtained many detrital zircons from the so-called green sandstone layer in the Barberton Greenstone Belt of South Africa, where numerous Hadean to Mesoarchean detrital zircon grains have already been found.

They performed paleointensity measurements on these grains, and those which produced acceptable intensities; they did follow up petrography via SEM (BSE and CL, and of course EDS), to find magnetite inclusions, followed by U-Pb geochronology via SHRIMP. This methodology is, of

course, sound, and is similar to studies already published from the Jack Hills area of western Australia, where the authors have already worked and published data from. The paleo-intensity measurements appear to be near identical between the two sites, and are apparently constant between ~ 3.9 Ga and ~ 3.4 Ga, suggesting these two pieces of the crust did not move significantly along an apparent polar wander path, which appears to invalidate having plate tectonics, but suggesting instead that stagnant lid tectonics was dominant at this time.

Originality and significance. The results are indeed significant; if true, it would allow us to infer that plate tectonics was not operating before the Mesoarchean. This is much debated in the geological community, with estimates for when plate tectonics started varying between the end of the Proterozoic and the Hadean. Most scientists usually estimate either a Meso- or Neoproterozoic onset, but even this remains moot. This, of course, will have implications for life on our planet and others. As for novelty, this is less certain; this methodology has been presented before and has been vigorously debated in the geological community. Hence the authors need to find primary magnetite inclusions, as well as consistent ages throughout a single grain. Here, the only thing that has changed is from talking about the existence of a magnetic field around the Earth and the state of the core, to plate tectonics itself.

Data and methodology. The methodology is sound, as stated above. It is a valid approach, showing the paleointensity values in the zircons, the existence of magnetite inclusions in them, and their documentation by BSE, EDS and CL, coupled with U-Pb geochronology. The devil, however, is in the detail. There are not enough samples and data to form a solid conclusion, which leads to the appropriate use of statistics and treatment of uncertainties.

I am no specialist in paleointensity, but I can assess this: U-Pb geochronology. The SHRIMP U-Pb shows (approximately) 76 analyses from 35 grains. Zircons grains are notorious heterogeneous in terms of age (Pb loss), hence the advantages of spot dating of various zones and the documentation by BSE and CL. Therefore, it is important to document this heterogeneity by imaging and age dating. For statistics, a good number of analyses, therefore, is between three and five, with five being preferable. This means that of the 35 grains in this study, only seven have enough analyses. Of these seven:

BGS1z9 has 13 analyses (13 concordant), with concordant $^{207}\text{Pb}/^{206}\text{Pb}$ ages between 3804 Ma and 4115, a time of over 300 Myr, not suitable.

BGS1z11 has eight analyses (eight concordant), with concordant $^{207}\text{Pb}/^{207}\text{Pb}$ ages between 3721 Ma and 3784 Ma, a time of less than 100 Myr, suitable.

BGS2z2 has seven analyses (seven concordant), with concordant $^{207}\text{Pb}/^{206}\text{Pb}$ ages between 4017 Ma and 3468 Ma, a time of over 500 Myr, not suitable.

BGS2z10 has six analyses (six concordant), with concordant $^{207}\text{Pb}/^{206}\text{Pb}$ ages between 3226 Ma and 4056 Ma, a time over 700 Myr, unsuitable.

BGS2z14 has three analyses (three concordant), with concordant $^{207}\text{Pb}/^{206}\text{Pb}$ ages between 3748

Ma and 3907 Ma, a time over 100 Myr, not suitable.

BGS2z15 has three analyses (three concordant), with concordant $^{207}\text{Pb}/^{206}\text{Pb}$ ages between 3388 Ma and 3769 Ma. A time of almost 400 Myr, is not suitable.

BGS5z12 has seven analyses (six concordant), with concordant $^{207}\text{Pb}/^{206}\text{Pb}$ ages between 3691 Ma and 3782 Ma, a time of less than 100 Myr, suitable.

Therefore, in my opinion, of the seven grains with good paleointensity, and sufficient analyses, only two are actually suitable. Additionally, if we assume a 100 Myr-age bin, both present a snapshot of only the 3700-3800 M age bin. You will still need an additional zircon to have an MSWD Reduced chi-squared statistic), for the one age bin, and I actually recommend an additional two age bins, preferably from more than one location. The authors have also published such data from Jack Hills and used it in this study, but I did not evaluate that study herein. So, on this, I find this study unsuitable and far-fetched.

In conclusion, it is actually very difficult to evaluate this study. There are several other geological points to consider: there is a lot of geological debate around the existence of plate tectonics against stagnant lid tectonics, and what the evidence is for and against it. The authors completely ignore this. See some references below. Paleomagnetic data does not exist alone, this needs to be considered. It is easy to refer the authors to the papers of Nutman and Friend on the subject. Additionally, these are detrital data, so in a sense, they are not 'in-situ'. Therefore, trying to evaluate paleo-latitude in my mind is a bit of nonsense. These zircons come from blocks which do not exist any longer. The oldest evidence of stable crust is from 3.1 Ga in the case of the Kaapvaal Craton, and 2.7 Ga in the case of the Yilgarn Craton.

These greenstone belts on them are not formed on the stable crust, and we have no knowledge about how 'exotic' they may be, and what the basement truly was. They could be nappes transported over large distances, for example. Any existence of the movement of the blocks needs to be considered, and actually, there is no good evidence. There is a hypothetical 'superblock' called Isukasia documented mostly based on 3.6 Ga magmatism and metamorphism worldwide. How do the authors account for such widespread magmatic and metamorphic ages without discussion? Paleomagnetic studies also account for so-called van der Voo criteria, especially in the Precambrian. Paleointensity in this study seems to try to bypass all this, but actually, it is still relevant. Adequate numbers of sites, field tests, rock magnetic studies on the zircon grains, cooling ages, and ages of magnetism. All of this seems to be ignored in the quest for a good story.

Suggested improvements. Please consider the advice above, generate at least three different age bins between 3.0 Ga and 4.1 Ga, with between three and five suitable zircons. These zircons have to have good paleo-intensity measurements, documentation by SEM in BSE, EDS and CL of magnetite grains, and with between three and five U-Pb geochronology ages that are concordant (preferably more than 95% concordant), with ages falling within this 100 Myr age window. Consider the van der Voo criteria and the geology. This is a huge task, I know, but if you are going to make huge, revolutionary and ground-breaking research, this is what is needed; there is no other way.

References. Generally sufficient, although almost all of this type of work is mostly done only by this

research team. However, some more citations of the limitations, i.e., Evans (2018) - RESEARCH FOCUS: Probing the complexities of magnetism in zircons from Jack Hills, Australia. *Geology* 46, 479-480, would be good, or Nutman et al. (2021) - Fifty years of the Eoarchean and the case for evolving uniformitarianism. *Precambrian Research* 367, 106442.

Clarity and context. Generally, very good, but as I said, the devil is in the detail. One can present all the supportive statistics one wants, but you cannot ignore the fact: there is not enough data from enough zircons even to begin to consider your case. Additionally, to disprove plate tectonics in the Archean, one has to look further and deeper in the literature.

I conclude with a quote: Deciphering the Eoarchean to Paleoarchean geological record must come from observation, not modelling.

Response to Reviewer Comments

Reviewer 1:

Comment: “I can follow the line of arguments developed in the study. The interpretation is reasonable and might stand the test of time. The ‘plate tectonic test’ (to make it highly unlikely that plate motion would have occurred during 3.9-3.4 Ga) is performed with Phanerozoic plate motion models. This is logical since plate tectonics for sure was active during the entire Phanerozoic.”

Response: We thank the reviewer for this assessment.

Comment: “However, the BGS and JH sample locations are associated with the Kaapvaal and Jilgarn-Pilbara cratons for which the oldest APWP has been proposed for a period between ~ 3.1 and ~ 2.7 Ga: they would be juxtaposed and constitute the Vaalbara super-craton (e.g. Zegers et al., 1998; de Kock et al., 2009). OK, this is younger than the time span considered in the present study and the APWP is not beyond discussion (e.g. Evans and Muxworthy, 2019). Some discussion on how the paleolatitudes compare and what would be the paleolatitude trajectory of the Vaalbara super-craton between ~ 3.1 and ~ 2.7 Ga would add. Plate motion of several cm per year to over 1 m per year have been argued for (the latter is very high for present-day standards). Is the change over from stagnant-lid tectonics to (present-day-style?) plate tectonics occurring within a few hundreds of millions of years?”

Response: We agree with the reviewer that the previous work on the hypothesized Vaalbara super-craton is not beyond discussion. As is also clear in the papers cited by the reviewer (specifically Zegers et al., 1998, and de Kock et al., 2009, v s. Evans and Muxworthy, 2019), there is debate on the existence of a single craton involving the Kaapvaal and Pilbara. The involvement of the Yilgarn craton is another issue of debate; geologic ties between the Pilbara and Yilgarn for these ages are not straightforward. However, we are comfortable with the close geological association marked by the Ventersdorp Supergroup (Kaapvaal craton) and Fortesque Group (Pilbara craton), but these date to 2.7-2.8 Ga. Thus, there is a 700 million-year-long age gap between this association and the time when our Jack Hills and Barberton paleointensity/paleolatitude records start to diverge (~ 3.4 Ga). This duration is roughly equivalent to that we examine to constrain the median maximum latitudinal motion characteristic of modern plate tectonics. Given that the median maximum latitudinal motion of 0-600 Ma plates is $\sim 76^\circ$, it follows that even very large paleolatitude differences between the hypothesized ‘Vaalbara’ super-craton at 2.7-2.8 Ga, and the ~ 3.4 Ga JH/BGS zircons could be reconciled by plate motion rates comparable to those of modern plate tectonics. We have added a paragraph in a new section 6 to the supplement with the references noted above, and discuss this issue for context. However, we prefer to restrict our main text to the test of moving vs. stationary lithosphere.

Comment: “What if present-day NW Australia and South Africa were juxtaposed to each other already before ~ 3.1 Ga? OK, they would be stationary (latitudinally) based on the paleointensity results presented

here, but is it then warranted to infer a global signal?”

Response: There are differences in the Barberton zircons recorded by geochemistry, referenced in our original submission, that suggest more than one plate source. This interpretation, compared with predictions from 0-600 Ma plate motions, leads us to infer we are seeing a global signature.

Comment: “What are thoughts on when plate tectonics (implicitly present-day style?) would have initiated?”

Response: The start of a divergence of the Jack Hills and Barberton records could be the start of plate tectonics as defined by large-scale horizontal motions. Although not explicitly the focus of our manuscript, we feel this is not an unreasonable inference, and we have now added it. We thank the reviewer for stimulating our thoughts on this!

Below are specific comments.

Comment: “First paragraph. With the onset of plate tectonics present-day style plate tectonics is implied? In the stagnant-lid regime there was volcanism as well? So, nutrient supply (not per se recycling) was possible?”

Response: We do not exclude nutrient cycling or volcanism. We are explicitly saying that these styles must have been different from modern plate tectonics, which is typified by very large horizontal motions over >500 million-year-long durations.

Comment: “Main text. P2 first paragraph. I can follow very well the line of reasoning developed but there is quite some generic paleomagnetic knowledge assumed implicitly. That the field is dipolar and that inclination and intensity thus scale with latitude. This probably might be presumed knowledge. Does the same also pertain to that individual detrital grains in a sediment cannot provide directional information on paleolatitude (probably clear) but that paleointensity still can do? This might warrant a concise explanation.”

Response: This information is explicitly provided in the supplement. We appreciate the reviewer’s viewpoint, but we have had to make decisions on where to place detailed supporting content versus essentials for the non-specialist scientist. Because the second reviewer did not raise this, we feel that the description in the supplement is sufficient, but we nevertheless thank the reviewer for the suggestion.

Comment: “References 16 and 17 refer to much younger rocks than dealt with here?”

Response: Yes, but this is just an example to show what we know of the field from the oldest extant rocks that are not completely compromised by metamorphism.

Comment: “What is implied with ‘higher grades’? The younger sedimentary units have not been or barely

metamorphosed?”

Response: This refers to extant rocks older than 3.45 Ga not younger sedimentary rocks hosting older zircons. We now clarify the grade (amphibolite).

Comment: “Second paragraph. Zircon by itself cannot provide paleomagnetic information, it is the magnetite inclusions that are responsible for that.”

Response: Yes. Rewritten to emphasize that magnetic inclusions are the magnetic carrier.

Comment: “Third paragraph. That the inclusions have not been magnetically reset is crucial. Perhaps it is wise to briefly recapitulate why the grains are not reset. Playing devil’s advocate: the reason for the non-varying paleointensity between 3.9 and 3.4 Ga is that the crystal radiometric ages vary between those two values, but the magnetic age would be 3.4 Ga for all of them.”

Response: We now include the multiple lines of evidence for the lack of resetting of the JH zircons as requested in a new supplementary section 1.0. We also expand in the SI on reasons the Barberton zircons are not reset.

Comment: “P3 top paragraph. The isolated iron particles are not consisting of metallic iron isn’t it? Be specific as to their magnetite nature I would say.”

Response: We had tried to be conservative, referring to Fe as being detected by EDS analysis. We see now that this could be interpreted as native iron, which was not our intent. Fe has been changed throughout to iron oxide or magnetite where appropriate.

Comment: Second paragraph and throughout text: recalculate emu to Am² since that is the proper SI unit.

Response: Done.

Comment: “The reason to adopt the 565 °C paleointensity protocol in the Jack Hills studies is to see through later medium grade metamorphism. Metamorphic temperatures that prevailed require laboratory heating to 565 °C for demagnetization and remagnetization experiments to obtain meaningful results. What is the discriminative power of the paleointensity at that temperature? The 565-580 °C portion of the blocking/unblocking temperature spectrum is small. In the Barberton Greenstone Belt lower metamorphic temperatures prevailed so that in principle a larger portion of the demagnetization spectrum can be used for meaningful partial thermoremanent magnetization acquisition experiments: how low a laboratory temperature is required to see through the metamorphism in the Barberton Greenstone Belt? Do the paleointensity results at let’s say 520 °C deliver the same outcome as those at 565 °C?”

Response: We thank the reviewer for raising this point. There are several reasons for the choice of 565 °C in this case. One is for comparison with the JH record which also used 565

°C. Another is to be well above the metamorphic temperature for our paleointensity estimate. The reviewer is correct in noting that there is a difference with the Jack Hills, which have experienced a higher metamorphic temperature. However, the other related reason – which we did not explicitly state but showed Figure 2b to illustrate– is to be within the unblocking of those grains most likely to preserve the primary paleointensity value. With single domain grains, we can expect unblocking closer to the Curie temperature, as compared to 520 °C. And Figure 2b shows a break in slope from a gradual demagnetization curve to a sharp drop in magnetic intensity; 565 °C is within this sharp drop and is better suited for a paleointensity estimate because we can have more confidence that this is within the single domain range. We now include this additional justification in the revised Methods.

Comment: “Third paragraph: the zircons with a much younger rim (how much is much?) presumably do not qualify. OK, the paleointensity is representative of the youngest age.”

Response: Yes, we assign the youngest age to the magnetization age. We have expanded on our exploration of different areas within the zircon for geochronology by describing the analyses conducted on each of the zircons in the revised supplement. We provide 77 additional age analyses to further bolster the age accuracy.

Comment: “P4 discussion. Perhaps it is wise to provide some more information on plate tectonics vs stagnant lid tectonics. Are these seen as the only two options? What is the difference between a mantle plume and a heat pipe?”

Response: There are multiple ways crust might be constructed by stagnant lid tectonics, but the specific test here is of a plate tectonic style of mobility. However, we now add an additional reference to a review paper (Nutman et al., 2021) that discusses salient issues. In typical usage, heat pipes describe transfer of heat by advection along narrow conduits in the lithosphere, together with downward advection of cold lithosphere. A mantle plume is generally thought to transfer heat from deeper in the mantle, and there is no substantial downward flow near the mantle plume location. We feel the references to Moore and Webb (2013) and Nutman et al. (2021) are sufficient to address this for the reader.

Comment: “P5 top. But just before the constant paleointensity has been used to argue for no (significant) plate motion and here it is related to geodynamo efficiency. How can somebody tell apart the difference? The reasoning in the final bits may be a little quickly developed.”

Response: Yes, we agree. We have added a new section to the supplement to provide context. During the last 200 myr, paleomagnetic data (directions and intensity) and dynamo models show variations that probably reflect changes in the pattern of core-mantle boundary heat flux that in turn influence dynamo efficiency. If there had been large changes in paleointensity between 3.4 and 3.9 Ga, the solution would be non-unique. We would be unable to separate changes in dynamo efficiency from plate motion. But we do not see such changes, and therefore we can make inferences on latitudinal stability.

Comment: “P5 sample preparation and selection. What is the reason for picking a NRM cut-off in the first place? Just practical issues? If the samples are not sufficiently magnetized they will deliver meaningless results, or is there yet another reason?”

Response: The cutoff is a practical issue of being able to measure accurately the NRM and the demagnetized value. NRMs less than the cutoff value can be measurable, but their demagnetized value could fall beneath the sensitivity of the ultra-high sensitivity SQUID magnetometer at the University of Rochester. We have added a line to the Methods to better explain this.

Comment: “P6 geochronology. Curiosity, where does the geochronological SQUID3 acronym stand for?”

Response: Technically SQUID3 stands for nothing, other than the name of the software. We now add this to the Methods. This is the name used in the literature; it is capitalized but it is not an acronym; instead it seems to be a play on words, following SHRIMP (which is an acronym) to follow a seafood theme! In any case, this is the common usage in the geochronology community.

Comment: “Figure 1 caption. Fe is not metallic Fe, name it magnetite or Fe-oxide?”

Response: Corrected.

Comment: “Figure 2. In case of zoned zircons, the youngest growth rim has magnetically reset all older portions of the respective grain? Zircon grows at higher temperatures than the maximum magnetite (un)blocking temperature?”

Response: Yes, this is our interpretation. We have further explained this in the revised Methods, and in the description of each grain.

Comment: “What are the demagnetization diagrams of the dated grains shown in c), f) and i)? In a) and b) (added vector difference sum decay curve?) the 565 °C value is almost 0.5 of the starting NRM, what is that for c), f) and i)? Show the age dating of BGZ5-z1?”

Response: In the 565 °C technique, magnetometer measurements are not made at intermediate temperatures to keep the total number of temperature treatments to a minimum, minimizing the risk of sample alteration. The NRM values for these grains, and NRM after treatment at 565 °C, are provided in Table 2, and of course will be available in the MagIC database. There is no significant trend in the NRM/NRM_{565} value with age (e.g. Pearson Correlation coefficient of -0.06). However, because the NRM/NRM_{565} value can be influenced by the directional characteristics when detrital zircons are the subject of study (i.e., varying angles between a low temperature overprint and a high temperature primary component), we do not advocate it as an indicator of rock magnetism. Instead, we feel the multidomain checks, also summarized in Table 2 and are all at low values, are better indicators of the viability of the

rock magnetic recorders.

Comment: “Figure 3. Why are the other zircon paleointensity data shown in grey? Are they from yet another region? They seem to be ‘high-ish’. How are the uncertainty ranges in the paleointensity calculated? 1 sigma, 2 sigma? It seems that the means of binned intervals do not come with an uncertainty, yet in extended data table 4 most bins contain more than a single value.”

Response: The grey data, identified in the caption, are not zircon data but are paleointensities from other single crystals from extant whole rocks. We now explicitly state they are from extant whole rocks. We had previously discussed these results relative to the zircon results at the end of Supplementary section 3.0 (now 4.0) and for convenience repeat that text here: “Paleointensities from the SCP analyses of Nondweni dacite samples are within the value of BGS values of the same age, but SCP of Barberton dacites at 3.45 Ga are higher. Both the Nondweni and Barberton SCP values are plotted as individual results from two relatively shallow intrusions and these might be expected to sample higher frequency variations of the geomagnetic field than the zircons.” The uncertainty assignments on the individual zircon results follow the empirical approach used in Tarduno et al. (2015, 2020), based on the comparison of full Thellier results and 565 °C results. Uncertainties of the combined BGS/JH data set are standard errors and are identified in the figure caption of Figure 4. We do not plot the uncertainties of the individual age bins in some other figures because that would obscure other data, but as noted these standard errors are available in Extended Data Table 4.

Comment: “Extended data figures 1-3. Put the respective zircon grain labels on the appropriate panels.”

Response: Done.

Comment: “P12. Extended data figure 2. It is not metallic Fe inclusions. Extended data figure 4. Here paleointensity data are plotted without uncertainty intervals.”

Response: We have modified the figure to read Fe-oxide. We feel it is better to plot the data without the uncertainties to see the trends and fitting. The uncertainties are plotted in the main text figure.

Comment: “Extended data figure 5. What is the red diamond in panel b?”

Response: The point where the maximum latitudinal distance is reached. This was accidentally omitted and has been added. We thank the reviewer for catching this.

Comment: “Extended data table 2. Delta565 is the angle between the TRM vector and the applied field. It shows values up to 29°. Ideally it should be close to zero. What is the underlying reason for the measured deviations? The amount of MD effects seems very low.”

Response: The deviations could have several sources. The rock magnetic source could be an anisotropy of the collection of magnetic particles in a given zircon. However, these are extremely challenging experiments and we can not exclude small error contributions related to the alignment of the laser beam and a zircon. We have noted this in the revised text.

Comment: “Supporting information. P3. Third paragraph. What is the meaning of acronym PDM? Check other acronyms throughout the document.”

Response: Thank you for catching this. Paleomagnetic dipole moment. We have noted this in the revised manuscript.

Comment: “P5. Third paragraph. Presumably the likelihood of observing two sites with no motion on separate plates is low. Is that written as such?”

Response: Yes. We have clarified the sentence and thank the reviewer for catching this.

Reviewer 2

Comment: “Summary. The manuscript submitted by Tarduno et al. (2022-05-07225; Hadean to Paleoproterozoic stagnant lid tectonics recorded by the paleomagnetism of single zircons of South Africa and Australia) aims to use paleointensity data from Hadean- to Mesoarchean-age single zircons to infer that plate/mobile lid tectonics (as we know it) was not operating at this time. Instead, they argue that stagnant lid tectonics was operational. To do this, they obtained many detrital zircons from the so-called green sandstone layer in the Barberton Greenstone Belt of South Africa, where numerous Hadean to Mesoarchean detrital zircon grains have already been found. They performed paleointensity measurements on these grains, and those which produced acceptable intensities; they did follow up petrography via SEM (BSE and CL, and of course EDS), to find magnetite inclusions, followed by U-Pb geochronology via SHRIMP. This methodology is, of course, sound, and is similar to studies already published from the Jack Hills area of western Australia, where the authors have already worked and published data from. The paleo-intensity measurements appear to be near identical between the two sites, and are apparently constant between ~ 3.9 Ga and ~ 3.4 Ga, suggesting these two pieces of the crust did not move significantly along an apparent polar wander path, which appears to invalidate having plate tectonics, but suggesting instead that stagnant lid tectonics was dominant at this time.”

“Originality and significance. The results are indeed significant; if true, it would allow us to infer that plate tectonics was not operating before the Mesoarchean. This is much debated in the geological community, with estimates for when plate tectonics started varying between the end of the Proterozoic and the Hadean. Most scientists usually estimate either a Meso- or Neoproterozoic onset, but even this remains moot. This, of course, will have implications for life on our planet and others.

Response: We thank the reviewer for this assessment.

Comment: “As for novelty, this is less certain; this methodology has been presented before and has been vigorously debated in the geological community... Hence the authors need to find primary magnetite inclusions, as well as consistent ages throughout a single grain. Here, the only thing that has changed is from talking about the existence of a magnetic field around the Earth and the state of the core, to plate tectonics itself. ”

Response: We note that our work is the first to make conclusions on mobile versus non-mobile lithosphere based on data constraining latitudinal motion for the Eoarchean to Paleoproterozoic interval. As discussed below and further examined with new data, we have probed different crystal areas within a zircon and obtain weighted mean ages. When a younger rim is present, we equate its age with the time of magnetization.

Comment: “Data and methodology. The methodology is sound, as stated above. It is a valid approach, showing the paleointensity values in the zircons, the existence of magnetite inclusions in them, and their documentation by BSE, EDS and CL, coupled with U-Pb geochronology. The devil, however, is in the detail. There are not enough samples and data to form a solid conclusion, which leads to the appropriate use of statistics and treatment of uncertainties.”

Response: We thank the reviewer for this assessment of our use of statistics. As detailed

below, we feel our sample set is sufficient to justify our conclusions.

Comment “I am no specialist in paleointensity, but I can assess this: U-Pb geochronology. The SHRIMP U-Pb shows (approximately) 76 analyses from 35 grains. Zircons grains are notorious heterogeneous in terms of age (Pb loss), hence the advantages of spot dating of various zones and the documentation by BSE and CL. Therefore, it is important to document this heterogeneity by imaging and age dating. For statistics, a good number of analyses, therefore, is between three and five, with five being preferable.”

“This means that of the 35 grains in this study, only seven have enough analyses. Of these seven: BGS1z9 has 13 analyses (13 concordant), with concordant 207Pb/206Pb ages between 3804 Ma and 4115, a time of over 300 Myr, not suitable. BGS1z11 has eight analyses (eight concordant), with concordant 207Pb/207Pb ages between 3721 Ma and 3784 Ma, a time of less than 100 Myr, suitable. BGS2z2 has seven analyses (seven concordant), with concordant 207Pb/206Pb ages between 4017 Ma and 3468 Ma, a time of over 500 Myr, not suitable. BGS2z10 has six analyses (six concordant), with concordant 207Pb/206Pb ages between 3226 Ma and 4056 Ma, a time over 700 Myr, unsuitable. BGS2z14 has three analyses (three concordant), with concordant 207Pb/206Pb ages between 3748 Ma and 3907 Ma, a time over 100 Myr, not suitable. BGS2z15 has three analyses (three concordant), with concordant 207Pb/206P ages between 3388 Ma and 3769 Ma. A time of almost 400 Myr, is not suitable. BGS5z12 has seven analyses (six concordant), with concordant 207Pb/206Pb ages between 3691 Ma and 3782 Ma, a time of less than 100 Myr, suitable.”

Therefore, in my opinion, of the seven grains with good paleointensity, and sufficient analyses, only two are actually suitable. Additionally, if we assume a 100 Myr-age bin, both present a snapshot of only the 3700-3800 M age bin. You will still need an additional zircon to have an MSWD Reduced chi-squared statistic), for the one age bin, and I actually recommend an additional two age bins, preferably from more than one location. The authors have also published such data from Jack Hills and used it in this study, but I did not evaluate that study herein. So, on this, I find this study unsuitable and far-fetched.”

Response: We thank the reviewer for raising this issue which led to our recognition of an error in our supplementary data table. With this correction, one of the large age ranges discussed by the reviewer is removed. However, the larger point is that we have probed different areas within a crystal to obtain weighted mean ages. When a younger “rim” is identified, we assign that to the magnetization age. However, we recognize that we did not describe this fully, and we did not have multiple age analyses for the younger grains. To address these issues, we have made the following additional measurements and associated revisions:

- *We now more clearly state in the manuscript that when a younger “rim” is identified, we assign that to the age of magnetization.*

- *We have added 77 additional age analyses, added to Table 1, and now report weighted means on nearly all zircons (there are a few exceptions described, but these do not affect our*

conclusions).

- *We have expanded the Methods, and in the Supplement we now include a description of the age analyses and assignment for each zircon. We note that our assumption that a single analysis would be sufficient to represent the age of the younger zircons is confirmed by our new data, but the additional analyses add rigor.*

Comment: “In conclusion, it is actually very difficult to evaluate this study. There are several other geological points to consider: there is a lot of geological debate around the existence of plate tectonics against stagnant lid tectonics, and what the evidence is for and against it. The authors completely ignore this. See some references below. Paleomagnetic data does not exist alone, this needs to be considered. It is easy to refer the authors to the papers of Nutman and Friend on the subject. ”

Response: Our first three citations discuss the geologic debate. However, as discussed below, we now specifically reference the review paper by Nutman et al. (2021) to further alert readers to the geologic data and interpretations.

Comment: “Additionally, these are detrital data, so in a sense, they are not ‘in-situ’. Therefore, trying to evaluate paleo-latitude in my mind is a bit of nonsense. These zircons come from blocks which do not exist any longer. The oldest evidence of stable crust is from 3.1 Ga in the case of the Kaapvaal Craton, and 2.7 Ga in the case of the Yilgarn Craton. These greenstone belts on them are not formed on the stable crust, and we have no knowledge about how ‘exotic’ they may be, and what the basement truly was. They could be nappes transported over large distances, for example.”

Response: If the zircons come from highly varied sources, rather than 2 more or less restricted areas, this further bolsters our interpretation because if more areas of the globe have been sampled, the prediction of plate tectonics is that they should have greater latitudinal variation.

Comment: “Any existence of the movement of the blocks needs to be considered, and actually, there is no good evidence. There is a hypothetical ‘superblock’ called Isukasia documented mostly based on 3.6 Ga magmatism and metamorphism worldwide. How do the authors account for such widespread magmatic and metamorphic ages without discussion?”

Response: In our manuscript we are of course not excluding the growth of continental crust during the Hadean to Eoarchean. We are saying that our data do not point to plate mobility typical of modern plate tectonics. We have emphasized this point by referencing the Nutman et al. review paper.

Comment: “Paleomagnetic studies also account for so-called van der Voo criteria, especially in the Precambrian. Paleointensity in this study seems to try to bypass all this, but actually, it is still relevant. Adequate numbers of sites, field tests, rock magnetic studies on the zircon grains, cooling ages, and ages of magnetism. All of this seems to be ignored in the quest for a good story. Suggested improvements. Please consider the advice above, generate at least three different age bins between 3.0 Ga and 4.1 Ga, with between three

and five suitable zircons. These zircons have to have good paleo-intensity measurements, documentation by SEM in BSE, EDS and CL of magnetite grains, and with between three and five U-Pb geochronology ages that are concordant (preferably more than 95% concordant), with ages falling within this 100 Myr age window. Consider the van der Voo criteria and the geology. This is a huge task, I know, but if you are going to make huge, revolutionary and ground-breaking research, this is what is needed; there is no other way.”

Response: The Van der Voo criteria apply to paleomagnetic directions, so we have developed a different set of criteria that leads us to select only ~3.5% of the zircons initially separated. These are very strict criteria, and are summarized in the Methods together with the success statistics. These criteria start with light microscopy, and extend to several different types of paleointensity data checks, to SEM analyses and finally U-Pb analyses. Field tests have already been published (e.g. Usui, Tarduno et al., 2009). Ultimately, the time signature of paleointensities from zircons at different global localities is important, and this is the first study to make this comparison.

As noted above, for the Eoarchean zircons our ages are based of weighted means of several spots. We have now better documented this. We have returned to the mount and added 77 new age analyses. The new ages are now weighted means based on at least 3 analyses as requested by the reviewer and are supplied in Table 1 (there are a few exceptions, explained in the Supplement). All the values and related statistics for paleointensity and inferred paleolatitude have been recalculated. Changes are quite minor and all of the conclusions we draw in the original manuscript are unchanged. However, we appreciate the reviewer’s insistence on further tests of the accuracy of our ages.

Comment: “References. Generally sufficient, although almost all of this type of work is mostly done only by this research team. However, some more citations of the limitations, i.e., Evans (2018) - RESEARCH FOCUS: Probing the complexities of magnetism in zircons from Jack Hills, Australia. *Geology* 46, 479-480, would be good, or Nutman et al. (2021) - Fifty years of the Eoarchean and the case for evolving uniformitarianism. *Precambrian Research* 367, 106442.”

Response: We are happy to add the Nutman et al. 2021 reference and to refer to it in the main text.

Comment: “Clarity and context. Generally, very good, but as I said, the devil is in the detail. One can present all the supportive statistics one wants, but you cannot ignore the fact: there is not enough data from enough zircons even to begin to consider your case. Additionally, to disprove plate tectonics in the Archean, one has to look further and deeper in the literature. I conclude with a quote: Deciphering the Eoarchean to Paleoproterozoic geological record must come from observation, not modelling.”

Response: We agree with the reviewer’s closing quote. We are not attempting to disprove plate tectonics on geological grounds. We are simply conducting a test of latitudinal mobility for the first time based on data, not modeling. By referring to the geologic work documenting shortening in the main text, we feel our revision now further emphasizes geologic data (also discussed in our first three references/citations), and we thank the reviewer for the suggestion.

Reviewer Reports on the First Revision:

Referee #1:

Authors have considered most of the points raised by me and worked them accordingly into the revised version; I am happy with that. Their choice of placing the longer argumentation points in the supplementary information rather than the main text is probably optimal. My core expertise is not in radiometric dating, but I note that the new ages seem to place the line of reasoning on a stronger footing. However, there remains an issue that I did not bring up before, apologies for that: the argument revolves around (the absence of) latitudinal motion as argued for, but longitudinal motion remains unconstrained because paleomagnetism (and paleointensity in the present case) cannot constrain that. This caveat is only touched upon kind of implicitly.

Hypothetically a crustal fragment can move around the globe while remaining at the same latitude: then paleolatitude and paleointensity would not change. Motion with a large latitudinal component is convincingly excluded but what about dominantly longitudinal motion? In absence of large latitudinal motion, large longitudinal motion seems unlikely but by necessity only one 'plate pair' can be assessed here (no other areas of similar old ages exist). The presence of a NS oriented subduction zone at the time would be highly coincidental, but can it be excluded or made unlikely based on geological arguments? Can the analysis on Phanerozoic plate motion tell us something along such lines?

[Additional comments solicited in response to referee #2's second review]

The point authors trying to make is that the zircons picked for their measurements contain primary inclusions, i.e. those not visually associated with cracks and the like. Further they pick single domain to vortex magnetite particles to be included in their final analysis. Here it comes: the highest temperature demagnetization end of those magnetites preserves a record of ancient TRM (induced when the youngest rim of the zircon was cooled through their blocking temperature spectrum) independent of their subsequent metamorphic history up to medium grade. The time-temperature diagrams of Pullaiah et al. 1975 EPSL (or equivalent newer studies) show that e.g. 100 Myr at 450 °C is equivalent to a few seconds at 520 °C.

So, higher unblocking temperatures see through the metamorphism. The host rock can be altered but the 565-580 °C segment survives that, but only in single domain or vortex particles. Because the particles are in a conglomerate the orientation is meaningless and authors must rely on paleointensity. It is comparable to ancient pottery shards that are not in situ: paleodirection of such shards is meaningless but paleointensity is very informative and is included in archeomagnetic data bases if the age of the shard is known. The 565-580 °C segment has only a very weak magnetic signal and only the Rochester paleomagnetic laboratory can faithfully measure such low magnetic moments (their small-bore SQUID magnetometer is an order of magnitude more sensitive than 'regular' 2G SQUID magnetometers. No other lab world-wide has the same set-up which brings the Rochester group in a unique position.

Referee #2:

I have now read the manuscript a second time and find that the authors have addressed many of my comments sufficiently. However, I have many lingering doubts in my mind. I have had the opportunity to read more about this subject, and with that, I feel that further doubts have been raised in my mind. This type of work is highly controversial and I see there is a lot of back and forth arguments and counter arguments, and there is certainly a lot of critique and contradictory data.

1. The documentation of the zircons, their ages, their inclusions, and their paleointensity alone is insufficient to conclusively document the primary nature of the paleomagnetic measurements. I refer the authors to the work of Weiss et al. (2015), who show that the rock units from Yilgarn containing Hadean zircons are extensively re-magnetised. Additionally, the authors cite Biggen et al. (2011) as providing evidence that the Onverwacht Group of the Barberton Greenstone Belt was not pervasively re-magnetised. However, from observation of these rocks in the field, they are highly variable, from almost completely altered to essentially pristine, depending on the locality. Therefore, it would be ideal if the authors could show this in studies that firmly demonstrate that the rock hosts are pristine through paleomagnetic (rock magnetic) and petrographic studies. The presence of magnetite inclusions is not sufficient. Zircon could be an effective blocker of chemical alteration of magnetite while still being heated above the Curie point. However, it depends. I also draw the authors to the work of Fu et al. (2021), who stated that very old zircons from these rocks only had secondary magnetite etc., related to cracks within the zircon. I see similar studies from Barberton. Therefore, there is a lot of debate going back and forth on this topic...so making such statements on the geodynamo and tectonics at this stage is highly debatable. I think you have to clear our minds of any doubt first.

So, all in all, there is still significant doubt in my mind, and among many others that I see in the literature, to warrant this manuscript not being published, unfortunately, until they can further demonstrate, independently, that these rocks have not been re-magnetised at a later date. However, I suggest that further review may be able to help resolve this.

Author Rebuttals to First Revision:**Response to Reviewer Comments****Reviewer 1:**

Comment: “Authors have considered most of the points raised by me and worked them accordingly into the revised version; I am happy with that. Their choice of placing the longer argumentation points in the supplementary information rather than the main text is probably optimal. My core expertise is not in radiometric dating, but I note that the new ages seem to place the line of reasoning on a stronger footing. ”

Response: We thank the reviewer for this assessment.

Comment: “However, there remains an issue that I did not bring up before, apologies for that: the argument revolves around (the absence of) latitudinal motion as argued for, but longitudinal motion remains unconstrained because paleomagnetism (and paleointensity in the present case) cannot constrain that. This caveat is only touched upon kind of implicitly. Hypothetically a crustal fragment can move around the globe while remaining at the same latitude: then paleolatitude and paleointensity would not change. Motion with a large latitudinal component is convincingly excluded but what about dominantly longitudinal motion? In absence of large latitudinal motion, large longitudinal motion seems unlikely but by necessity only one ‘plate pair’ can be assessed here (no other areas of similar old ages exist). The presence of a NS oriented subduction zone at the time would be highly coincidental, but can it be excluded or made unlikely based on geological arguments? Can the analysis on Phanerozoic plate motion tell us something along such lines? ”

Response: Paleolongitudinal motion is implicitly addressed in our analysis, but we agree with the reviewer that we did not explicitly discuss this issue. We have added a description to the Main text and Supplementary Information (now called Methods 5.0). Cases where longitudinal motion is dominant are rare (<5%) and these have significant associated latitudinal motion (>40°), inconsistent with the paleolatitude history inferred from the zircons.

[Additional comments solicited in response to referee #2’s second review]

The point authors trying to make is that the zircons picked for their measurements contain primary inclusions, i.e. those not visually associated with cracks and the like. Further they pick single domain to vortex magnetite particles to be included in their final analysis. Here it comes: the highest temperature demagnetization end of those magnetites preserves a record of ancient TRM (induced when the youngest rim of the zircon was cooled through their blocking temperature spectrum) independent of their subsequent metamorphic history up to medium grade. The time-temperature diagrams of Pullaiah et al. 1975 EPSL (or equivalent newer studies) show that e.g. 100 Myr at 450 °C is equivalent to a few seconds at 520 °C. So, higher unblocking temperatures see through the metamorphism. The host rock can be altered but the 565-580 °C segment survives that, but only in single domain or vortex particles. Because the particles are in a conglomerate the orientation is meaningless and authors must rely on paleointensity. It is comparable to ancient pottery shards that are not in situ: paleodirection of such shards is meaningless but paleointensity is very informative and is included in archeomagnetic data bases if the age of the shard is known. The

565-580 °C segment has only a very weak magnetic signal and only the Rochester paleomagnetic laboratory can faithfully measure such low magnetic moments (their small-bore SQUID magnetometer is an order of magnitude more sensitive than ‘regular’ 2G SQUID magnetometers. No other lab world-wide has the same set-up which brings the Rochester group in a unique position.

Response: We thank the reviewer for this summary that nicely highlights salient points of our analyses and rationale for primary zircon magnetizations.

Reviewer 2:

Comment: I have now read the manuscript a second time and find that the authors have addressed many of my comments sufficiently.

Response: We thank the reviewer for this assessment.

Comment: However, I have many lingering doubts in my mind. I have had the opportunity to read more about this subject, and with that, I feel that further doubts have been raised in my mind. This type of work is highly controversial and I see there is a lot of back and forth arguments and counter arguments, and there is certainly a lot of critique and contradictory data.

1. The documentation of the zircons, their ages, their inclusions, and their paleointensity alone is insufficient to conclusively document the primary nature of the paleomagnetic measurements. I refer the authors to the work of Weiss et al. (2015), who show that the rock units from Yilgarn containing Hadean zircons are extensively re-magnetised. Additionally, the authors cite Biggen et al. (2011) as providing evidence that the Onverwacht Group of the Barberton Greenstone Belt was not pervasively re-magnetised. However, from observation of these rocks in the field, they are highly variable, from almost completely altered to essentially pristine, depending on the locality. Therefore, it would be ideal if the authors could show this in studies that firmly demonstrate that the rock hosts are pristine through paleomagnetic (rock magnetic) and petrographic studies. The presence of magnetite inclusions is not sufficient. Zircon could be an effective blocker of chemical alteration of magnetite while still being heated above the Curie point. However, it depends. I also draw the authors to the work of Fu et al. (2021), who stated that very old zircons from these rocks only had secondary magnetite etc., related to cracks within the zircon. I see similar studies from Barberton. Therefore, there is a lot of debate going back and forth on this topic...so making such statements on the geodynamo and tectonics at this stage is highly debatable. I think you have to clear our minds of any doubt first.

So, all in all, there is still significant doubt in my mind, and among many others that I see in the literature, to warrant this manuscript not being published, unfortunately, until they can further demonstrate, independently, that these rocks have not been re-magnetised at a later date. However, I suggest that further review may be able to help resolve this.

Response: We refer the reviewer to the excellent summary by Reviewer 1 that addresses this issue. We add the following points:

• *Our Supplementary Information Section 1.0 (now Methods 1.0, “Evidence for primary magnetite inclusions and magnetizations in JH zircons”) discusses most of these points, which we systematically addressed in Tarduno et al. (2020). We refer the reviewer to the Comment by Bono and Tarduno (2016) on the Weiss et al. (2015) paper, and to works by Dare et al. (2016) and Cottrell et al. (2016). These papers show that the analysis of Weiss et al. (2015) relied on an inappropriate use of statistics to define a putative remagnetization direction that is not present in the raw data, and that the unblocking temperature ranges identified by Weiss et al. (2015) are lower than those used to define the zircon characteristic magnetizations. The JH rocks have seen metamorphic temperatures of $\sim 475^\circ\text{C}$, but any magnetization isolated be-*

low and near that range is just an expected metamorphic overprint.

- *Our main text is very clear and open in pointing out that another group has been trying to disprove our results. A recent atom probe study by that group (Taylor et al., 2023) further illustrates the difference in magnetic recorders in our analyses and their own. In the Taylor et al. (2023) study, quantum diamond microscope (QDM) analyses from their prior work were used to identify a purported zone in a single JH zircon containing Fe particles, but these appear to be too small to be remanence carriers (cf. Tarduno et al., 2020). Taylor et al. (2023) assigned a maximum age of ~ 1.4 Ga to the zircon in this Fe-bearing zone. However, because this is younger than the minimum age of the zircon magnetization constrained by zircon microconglomerate tests (i.e., ~ 3 Ga, the depositional age of the conglomerate), this cannot be the high unblocking temperature remanence studied by Tarduno et al. (2015, 2020). We refer the reviewer to Tarduno et al. (2020) where it is explained that QDM data are not measurements of the actual magnetic remanence (the QDM cannot measure these low intensity signals, cf. Tarduno et al., 2020), but instead the measurement of a laboratory induced magnetization that can enhance signals from Fe-oxides that are not carriers of the primary signal. Thus, while the atom probe work is just on one zircon, the results do serve to illustrate differences between secondary signals highlighted in other papers and the primary recorders we have studied. We have added a note to Methods 1.0 to emphasize this difference.*

- *There is no evidence of heating above the Curie temperature in the BGS, and the metamorphic grade of these units is well constrained.*

- *Our Methods 1.0 section referenced the Fu et al. study (same group as referred to above) and why our light microscope, SEM, and EDS analyses demonstrate that the claim by Fu et al. that BGS zircons “contain virtually no ferromagnetic minerals” is erroneous.*